# Best Subset Selection: Optimal Pursuit for Feature Selection and Elimination

Zhihan Zhu [1]  Yanhao Zhang [1]  Yong Xia [1]

## Abstract

This paper introduces two novel criteria: one for feature selection and another for feature elimination in the context of best subset selection, which is a benchmark problem in statistics and machine learning. From the perspective of optimization, we revisit the classical selection and elimination criteria in traditional best subset selection algorithms, revealing that these classical criteria capture only partial variations of the objective function after the entry or exit of features. By formulating and solving optimization subproblems for feature entry and exit exactly, new selection and elimination criteria are proposed, proved as the optimal decisions for the current entry-and-exit process compared to classical criteria. Replacing the classical selection and elimination criteria with the proposed ones generates a series of enhanced best subset selection algorithms. These generated algorithms not only preserve the theoretical properties of the original algorithms but also achieve significant meta-gains without increasing computational cost across various scenarios and evaluation metrics on multiple tasks such as compressed sensing and sparse regression.

## 1. Introduction

Subset selection is a classic topic in statistics and machine learning, with significant applications in feature selection (Kohavi & John, 1997; Das & Kempe, 2011), sparse regression (Miller, 2002; Das & Kempe, 2018), compressed sensing (Chen et al., 2001), maximum coverage (Feige, 1998) etc. Even in the era of large models, subset selection can effectively reduce training costs and enhance the instruction-following ability of large language models (LLMs) by selecting high-quality features (Wang et al., 2024).

The fundamental multivariate linear regression model with

---

[1]School of Mathematical Sciences, Beihang University, Beijing, China (Email: {zhihanzhu, yanhaozhang, yxia}@buaa.edu.cn). Correspondence to: Yong Xia <yxia@buaa.edu.cn>.

*Proceedings of the 42$^{nd}$ International Conference on Machine Learning*, Vancouver, Canada. PMLR 267, 2025. Copyright 2025 by the author(s).

coefficient vector $\boldsymbol{\beta} \in \mathbb{R}^{p \times 1}$ is expressed as follows:

$$\mathbf{y} = \mathbf{X}\boldsymbol{\beta} + \boldsymbol{\epsilon}, \qquad (1)$$

where $\mathbf{y} \in \mathbb{R}^{n \times 1}$ represents the response vector, $\mathbf{X} \in \mathbb{R}^{n \times p}$ is the design matrix, and $\boldsymbol{\epsilon} \in \mathbb{R}^{n \times 1}$ denotes the measurement noise. In subset selection, simpler models with fewer predictors are often favored. The goal is to select a subset of features by identifying nonzero coefficients (i.e., Active / Support Set $S$) in $\boldsymbol{\beta}$ that achieves a balance between accuracy and model simplicity. This leads to the best-subset selection problem, which can be formulated as follows:

$$\min_{\boldsymbol{\beta} \in \mathbb{R}^p} \mathcal{L}_n(\boldsymbol{\beta}) \triangleq \frac{1}{2n}\|\mathbf{y} - \mathbf{X}\boldsymbol{\beta}\|_2^2 \quad \text{s. t. } \|\boldsymbol{\beta}\|_0 \leq K, \quad (2)$$

where $K$ is maximum allowed sparsity level. Since problem (2) is NP-hard (Davis et al., 1997), significant efforts have been directed toward developing polynomial-time approximation algorithms (Dy & Brodley, 2000; Qian et al., 2015; Wei et al., 2015; Qian et al., 2017; Zhu et al., 2020).

Relaxation-based methods have been proposed to approximately solve (2) by replacing $\ell_0$ penalty with smooth approximations. Examples include Least Absolute Shrinkage and Selection Operator (LASSO) (Tibshirani, 1996), Adaptive LASSO (Zou, 2006), Smoothly Clipped Absolute Deviation (SCAD) (Fan & Li, 2001), Minimax Concave Penalty (MCP) (Zhang, 2010), etc. However, these methods could be computational burdensome (Hazimeh & Mazumder, 2020; Needell & Tropp, 2009) and are also difficult to control the number of selected features.

Another widely used class of methods is greedy algorithms, known for their high computational efficiency and simplicity. These methods perform subset selection directly by selecting and eliminating basis based on feature importance. Notably, the criteria for feature selection and elimination in this category are generally consistent, differing only in the underlying combination strategies.

**Correlation-based Selection.** Greedy algorithms typically select features based on their correlation with residuals, calculated as follows:

$$\mathbf{r}^k = \mathbf{y} - \mathbf{X}\boldsymbol{\beta}^{k-1}, \; j^* = \underset{j \in S^c}{\operatorname{argmax}} \frac{|\mathbf{r}^{k^T}\mathbf{X}_j|}{\|\mathbf{X}_j\|_2}, \qquad (3)$$

where $\mathbf{X}_j$ is the $j$-th column of $\mathbf{X}$, $S^c$ is the complement of support $S$, $\boldsymbol{\beta}^{k-1}$ denotes the updated coefficient on $S$, and

$\mathbf{r}^k$ represents the residual at step $k$. Representative methods include Matching Pursuit (MP) (Mallat & Zhang, 1993), Orthogonal Matching Pursuit (OMP) (Pati et al., 1993), CoSaMP (Needell & Tropp, 2009), Least Angle Regression (LARS) (Efron et al., 2004) and Adaptive Best-Subset Selection (ABESS) (Zhu et al., 2020), commonly used in compressed sensing and sparse regression.

**Wald-T Test Statistics-based Elimination.** Feature elimination often relies on the absolute value of Wald-T test statistics, defined as (here we assume the columns of $\mathbf{X}$ are centralized with zero mean for convenience):

$$|T_j| = \frac{|\boldsymbol{\beta}_j^{k-1}|}{M_{\boldsymbol{\beta}_j^{k-1}}}, \text{ where } M_{\boldsymbol{\beta}_j^{k-1}} = \frac{\|\mathbf{r}^k\|/\sqrt{df}}{\sqrt{\mathbf{X}_j^T \mathbf{X}_j}}, \; j \in S, \quad (4)$$

where $df$ serves as degree of freedom. Elimination is often based on minimizing the T-statistic or setting a threshold for deletion. Algorithms such as Iterative Hard Thresholding (IHT) (Blumensath & Davies, 2009), Hard Thresholding Pursuit (HTP) (Foucart, 2011), CoSaMP (Needell & Tropp, 2009) and Adaptive Best-Subset Selection (ABESS) (Zhu et al., 2020) employ this criterion to remove features.

In general, greedy methods can be regarded as a combination of correlation-based selection and T-statistic-based elimination, with different strategies integrated to perform subset selection. However, criteria (3) and (4), according to their formulas, focus solely on the individual significance of features, neglecting their interaction with other features. A feature that appears important within the current active set might become less significant when the active set changes, and conversely, a feature deemed less critical could gain importance under a different active set configuration. However, the current criteria fail to capture these dynamic properties.

In this paper, we revisit two classical criteria from an optimization perspective. By precisely modeling the selection and elimination subproblems, we reveal that the existing classical criteria (3) and (4) merely represent partial (one-step) variation of objective function after the entry or exit of features, arising from solving the subproblems by block coordinate descent method. Under this perspective, two novel importance criteria are proposed by solving the subproblems exactly. The new criteria consider both the significance of the features and their interactions with other features in the current active set simultaneously, with which the feature entry-exit strategies are proved as optimal decisions in the current subset selection process. Substituting the traditional criteria (3) and (4) with the proposed ones generates a series of enhanced best subset selection algorithms, which preserves their desirable theoretical properties (or even better) while achieving significant meta-gains across a variety of tasks, scenarios and evaluation metrics.

## 2. Revisit Two Classical Criteria

In this chapter, we revisit two classical criteria (3) (4) from an optimization perspective and develop a unified optimization model to uncover their fundamental characteristics.

The following viewpoints originate from ABESS (Zhu et al., 2020; 2022), which defined two types of sacrifices to measure the variation of objective $\mathcal{L}_n(\boldsymbol{\beta}) \triangleq \frac{1}{2n}\|\mathbf{y} - \mathbf{X}\boldsymbol{\beta}\|_2^2$:

1) Forward Sacrifice: For any $j \in S^c$, the magnitude of adding variable $j$ is defined as:

$$\eta_j = \mathcal{L}_n(\hat{\boldsymbol{\beta}}) - \mathcal{L}_n(\hat{\boldsymbol{\beta}} + \mathbf{D}^j\hat{\mathbf{t}})$$
$$= \frac{1}{2n}\left(\|\mathbf{y} - \mathbf{X}\hat{\boldsymbol{\beta}}\|_2^2 - \|\mathbf{y} - \mathbf{X}\hat{\boldsymbol{\beta}} - \mathbf{X}_j\hat{\mathbf{t}}_j\|_2^2\right)$$
$$= \frac{\mathbf{X}_j^T\mathbf{X}_j}{2n}\left(\frac{\hat{d}_j}{\mathbf{X}_j^T\mathbf{X}_j/n}\right)^2, \quad (5)$$

where $\mathbf{D}^j$ is defined as a diagonal matrix with $1$ at the $(j,j)$-th entry, while all other entries are $0$, $\hat{d} = \frac{\mathbf{X}^T(\mathbf{y} - \mathbf{X}\hat{\boldsymbol{\beta}})}{n}$, and $\hat{\mathbf{t}} = \operatorname{argmin}_{\mathbf{t}} \mathcal{L}_n(\hat{\boldsymbol{\beta}} + \mathbf{D}^j\mathbf{t})$. It evaluates which features outside active set are significant.

2) Backward Sacrifice: For any $j \in S$, the magnitude of removing variable $j$ is defined as:

$$\xi_j = \mathcal{L}_n(\hat{\boldsymbol{\beta}} - \mathbf{D}^j\hat{\boldsymbol{\beta}}) - \mathcal{L}_n(\hat{\boldsymbol{\beta}})$$
$$= \frac{1}{2n}\left(\|\mathbf{y} - \mathbf{X}\hat{\boldsymbol{\beta}} + \mathbf{X}_j\hat{\boldsymbol{\beta}}_j\|_2^2 - \|\mathbf{y} - \mathbf{X}\hat{\boldsymbol{\beta}}\|_2^2\right)$$
$$= \frac{\mathbf{X}_j^T\mathbf{X}_j}{2n}(\hat{\boldsymbol{\beta}}_j)^2, \quad (6)$$

which quantifies the irrelevant features in the support set.

Intuitively, for $j \in S$ (or $j \in S^c$), a large value of $\xi_j$ (or $\eta_j$) suggests that the $j$-th variable is potentially important. Formally, we demonstrate the connection between (5) (6) and (3) (4) in the following remark. This equivalence can be immediately verified by substituting $\mathbf{r}^k$ in $\hat{d}$.

***Remark*** 2.1. The criteria in (5) and (6) are strictly equivalent (up to a constant factor) to those in (3) and (4).

From an optimization standpoint, the existing criteria can be interpreted as the variation of objective function achieved by updating the support set with fixed coefficients in the first step of block coordinate descent. Specifically:

**Step 1 (Support update with fixed coefficient):** Maximize the correlation in (3) (or minimize the T-statistics in (4)) for support set updating is equivalent to solving (P0) (or (Q0)). The classical criteria (3) and (4) corresponds to the variation of objective function value in this part.

**Step 2 (Coefficient update with fixed support):** It is followed by refining the coefficients on the updated support set. This step also leads to a change in the function value, which, however, is not captured by the classical criteria.

**A. SUBSET SELECTION**

**A1.** The old correlation-based criterion corresponds to the partial decrement of the objective function

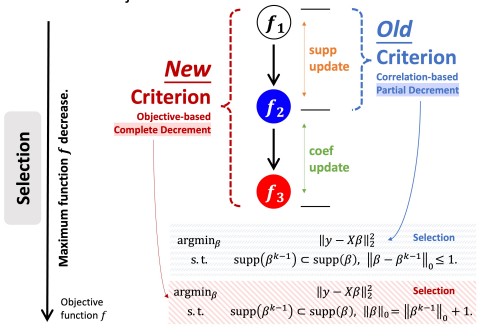

**A2.** The proposed new objective-based criterion corresponds to the complete decrement of the objective function

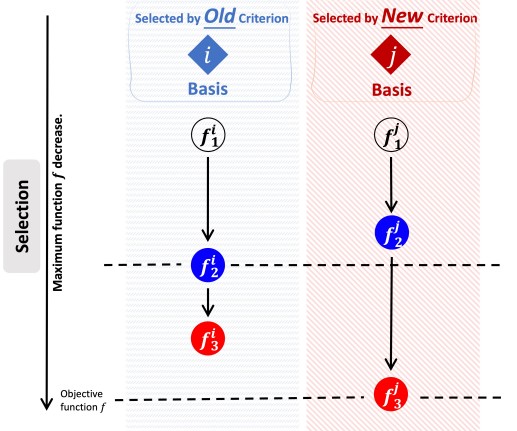

**B. SUBSET ELIMINATION**

**B1.** The old Wald-T statistics-based criterion corresponds to the partial increment of the objective function

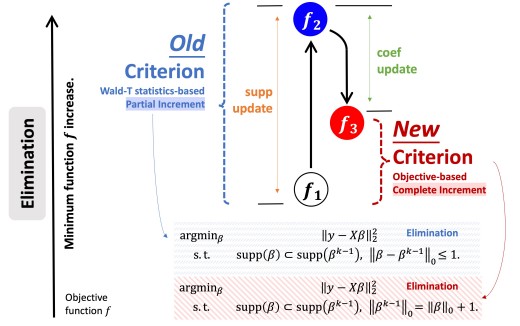

**B2.** The proposed new objective-based criterion corresponds to the complete increment of the objective function

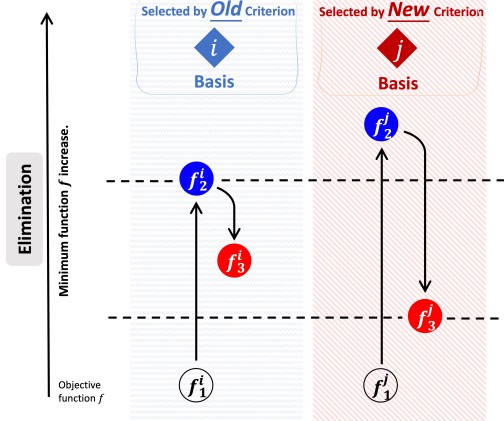

*Figure 1.* **A** (**B**) illustrates subset selection (elimination), with **A1** (**B1**) comparing objectives and optimization problems of the old and new criteria, and **A2** (**B2**) comparing function decreases (increases).

$$\operatorname*{argmin}_{\boldsymbol{\beta}} \quad \|\mathbf{y} - \mathbf{X}\boldsymbol{\beta}\|_2^2 \qquad\qquad (\text{P0})$$
$$\text{s. t.} \quad \|\boldsymbol{\beta} - \boldsymbol{\beta}^{k-1}\|_0 \le 1, \ \operatorname{supp}(\boldsymbol{\beta}^{k-1}) \subset \operatorname{supp}(\boldsymbol{\beta}).$$

$$\operatorname*{argmin}_{\boldsymbol{\beta}} \quad \|\mathbf{y} - \mathbf{X}\boldsymbol{\beta}\|_2^2 \qquad\qquad (\text{Q0})$$
$$\text{s. t.} \quad \|\boldsymbol{\beta} - \boldsymbol{\beta}^{k-1}\|_0 \le 1, \ \operatorname{supp}(\boldsymbol{\beta}) \subset \operatorname{supp}(\boldsymbol{\beta}^{k-1}).$$

As the model evolves to (P0) and (Q0), the limitations of classic criteria become more apparent. Specifically, it is observed that the constraints in (P0) and (Q0) (or equivalently, criteria (3) and (4)) only allows one change in the support set of $\boldsymbol{\beta}$ while the coefficients on the remaining support set are fixed. In the next step when coefficients are updated on newly selected support set, the influence of newly chosen feature on the remaining coefficients is not considered. The variation of function value in the first part measures the individual significance of the features, while the change in the second part assesses the interaction between features, where classical criteria fail to capture.

Therefore, the update strategy in (P0) and (Q0) (or criteria (3) and (4)) can be understood as the objective of performing one step of block coordinate descent, rather than an objective that takes into account the overall descent.

This issue will be addressed in the new model we develop in Section 3, which ensures an optimal solution at each step.

## 3. Optimal Selection and Elimination Statistics

### 3.1. Optimal Selection and Elimination Problem

As analyzed in Section 2, to obtain the optimal solution at each step, we consider the following optimization problems:

$$\operatorname*{argmin}_{\boldsymbol{\beta}} \quad \|\mathbf{y} - \mathbf{X}\boldsymbol{\beta}\|_2^2 \qquad\qquad (\text{P1})$$
$$\text{s. t.} \ \|\boldsymbol{\beta}\|_0 = \|\boldsymbol{\beta}^{k-1}\|_0 + 1, \ \operatorname{supp}(\boldsymbol{\beta}^{k-1}) \subset \operatorname{supp}(\boldsymbol{\beta}),$$

$$\operatorname*{argmin}_{\boldsymbol{\beta}} \quad \|\mathbf{y} - \mathbf{X}\boldsymbol{\beta}\|_2^2 \qquad\qquad (\text{Q1})$$
$$\text{s. t.} \ \|\boldsymbol{\beta}\|_0 = \|\boldsymbol{\beta}^{k-1}\|_0 - 1, \ \operatorname{supp}(\boldsymbol{\beta}) \subset \operatorname{supp}(\boldsymbol{\beta}^{k-1}),$$

where (P1) and (Q1) correspond to selection and elimination subproblems for each step exactly. Unlike the constraint in

(P0), the constraint in (P1) does not require the coefficients fixed on the remaining support set. Thus, compared to the incomplete descent obtained by one step of block coordinate descent with the classic criterion, the new optimization problem consider a complete descent (including both step 1&2), providing the optimal feature selection criterion. Same argument holds for the subset elimination problem in (Q1).

Figure 1 provides an intuitive comparison between the later-derived new criteria (from (P1) (Q1)) and the classic criteria (3) (4) (from (P0) (Q0)), highlighting their differences in terms of objectives. In the selection subproblem, the goal is to select a feature that maximizes the overall decrease in the objective function, as illustrated in **A1**. However, the classic criterion focuses solely on maximizing the immediate decrease after updating the support set, without accounting for the impact of subsequent coefficient updates. This limitation is evident in **A2**, where the feature $i$, selected by classic criterion, achieves the largest initial decrease $f_2^i$, but its subsequent updates result in less favorable outcomes.

Similarly, in the elimination subproblem, the objective is to eliminate a feature that minimizes the overall increase in the objective function. As shown in **B2**, the classic criterion selects basis $i$ because it results in the smallest immediate increase in the first step. However, after coefficient updates, the overall increase is much larger than that achieved by basis $j$, selected by new criterion.

### 3.2. Optimal Selection and Elimination Criteria

In this section, we solve (P1) and (Q1) to derive the optimal criteria for subset selection and elimination.

#### 3.2.1. OPTIMAL SELECTION CRITERION

We begin with deriving the optimal selection criterion.

**Lemma 3.1.** *Problem* (P1) *is equivalent to*

$$\underset{\boldsymbol{\beta}}{\arg\max} \quad \mathbf{y}^T \mathbf{X}_S \left( \mathbf{X}_S^T \mathbf{X}_S \right)^{-1} \mathbf{X}_S^T \mathbf{y} \qquad (P2)$$

$$\text{s. t.} \operatorname{supp}(\boldsymbol{\beta}^{k-1}) \subset S = \operatorname{supp}(\boldsymbol{\beta}), \; \|\boldsymbol{\beta}\|_0 = \|\boldsymbol{\beta}^{k-1}\|_0 + 1,$$

*where $S$ is the support set of $\boldsymbol{\beta}$, and $\mathbf{X}_S$ represents the columns of $\mathbf{X}$ corresponding to the subset $S$.*

The proof of Lemma 3.1 is detailed in Appendix A. Thus, problem (P1) is reformulated as (P2). A major challenge in solving (P2) is computing the inversion term $\left( \mathbf{X}_S^T \mathbf{X}_S \right)^{-1}$. To address this, we first introduce the following lemma.

**Lemma 3.2** (Forward Inverse). *Let $t = \alpha - \mathbf{v}^T \mathbf{A}^{-1} \mathbf{u} \neq 0$,*
$$\mathbf{B} = \begin{pmatrix} \mathbf{A}_{(n-1)\times(n-1)} & \mathbf{u} \\ \mathbf{v}^T & \alpha \end{pmatrix}_{n \times n}, \text{ where } \mathbf{A}^{-1} \text{ is known in}$$
*advance. Then the inverse of $\mathbf{B}$ is given by:*

$$\mathbf{B}^{-1} = \begin{pmatrix} \mathbf{A}^{-1} + \mathbf{A}^{-1}\mathbf{u}t^{-1}\mathbf{v}^T\mathbf{A}^{-1} & -\mathbf{A}^{-1}\mathbf{u}t^{-1} \\ -t^{-1}\mathbf{v}^T\mathbf{A}^{-1} & t^{-1} \end{pmatrix}. \qquad (7)$$

Lemma 3.2 can be obtained through simple matrix block operations. We can derive the following optimal feature selection criterion by applying Lemma 3.2.

**Theorem 3.3.** *Problem* (P2) *is equivalent (in the sense of identifying the true subset) to:*

$$\underset{j \in S_{k-1}^c}{\arg\max} \frac{\left( \mathbf{r}^{k^T} \mathbf{X}_j \right)^2}{\mathbf{X}_j^T \left( \mathbf{I} - \mathbf{X}_{S_{k-1}} \left( \mathbf{X}_{S_{k-1}}^T \mathbf{X}_{S_{k-1}} \right)^{-1} \mathbf{X}_{S_{k-1}}^T \right) \mathbf{X}_j}, \qquad (8)$$

*where $S_{k-1} = \operatorname{supp}(\boldsymbol{\beta}^{k-1})$.*

The proof of Theorem 3.3 is provided in Appendix B.

**Definition 3.4** (**Objective-based Selection**). By Theorem 3.3, the new criterion for feature importance outside the support set could be formulated as criterion (8).

**Remark** 3.5. When $\mathbf{X}_j$ is orthogonal to $\mathbf{X}_{S_{k-1}}$, the objective-based selection criterion (8) degenerates into the correlation-based criterion (3).

**Remark** 3.6 (Comprehensive Combination Effect). It is observed that the numerator of the objective-based selection criterion (8) is identical to that of the classic criterion (3). However, the denominator in (8) incorporates a projection matrix, which accounts for the interaction between the currently selected feature $j$ and the remaining features in $S_{k-1}$. Specifically, new criterion (8) can be interpreted as the correlation between $\mathbf{X}_j$ and $\mathbf{r}^k$ in the noise subspace of $\mathbf{X}_{S_{k-1}}$. As a result, the new criterion captures a comprehensive 'combination effect' in reducing the objective function.

**Remark** 3.7 (Computational Efficiency). Theorem 3.3 transforms the large matrix $\left( \mathbf{X}_S^T \mathbf{X}_S \right)^{-1}$ in (P2) into a smaller matrix $\left( \mathbf{X}_{S_{k-1}}^T \mathbf{X}_{S_{k-1}} \right)^{-1}$ in (8). In fact, when we update the coefficients on the support set $S_{k-1}$ at step $k-1$ using least squares $\left( \mathbf{X}_{S_{k-1}}^T \mathbf{X}_{S_{k-1}} \right)^{-1} \mathbf{X}_{S_{k-1}}^T \mathbf{y}$, this procedure involves either matrix inversion of $\mathbf{X}_{S_{k-1}}^T \mathbf{X}_{S_{k-1}}$ or solving the corresponding linear system via Cholesky decomposition. Notably, the inversion or Cholesky decomposition is required only once. Thus, despite the presence of an inversion term in the denominator of the new criterion (8), it does not incur additional magnitude of computational cost. We also perform runtime comparison between the algorithms using the classical criteria and the new criteria in Section 5, with only a slight difference, see Appendix O.

**Remark** 3.8. While writing this paper, we coincidentally came across the algorithm (SMP) with very similar criterion to (8) in (Tohidi et al., 2025). We independently derived the criterion with the completely different approaches taken. In that paper, a submodular perspective is used to derive a replacement algorithm for OMP, whereas our derivation is grounded in an unified optimization framework, with a specific focus on both optimal feature selection and elimination criteria. Moreover, in Section 3.3, we demonstrate that

our criteria can be applied as a meta-heuristic method to all heuristic best subset selection algorithms.

### 3.2.2. OPTIMAL ELIMINATION CRITERION

Similar to the selection problem, we make the following equivalent transformation to the original problem (Q1) in Lemma 3.9 and introduce Lemma 3.10 to assist in deriving the optimal elimination criterion.

**Lemma 3.9.** *Problem* (Q1) *is equivalent to*

$$\operatorname*{argmax}_{\boldsymbol{\beta}} \quad \mathbf{y}^T \mathbf{X}_S \left(\mathbf{X}_S^T \mathbf{X}_S\right)^{-1} \mathbf{X}_S^T \mathbf{y} \qquad (Q2)$$

s. t. $S = \operatorname{supp}(\boldsymbol{\beta}) \subset \operatorname{supp}(\boldsymbol{\beta}^{k-1})$, $\|\boldsymbol{\beta}\|_0 = \|\boldsymbol{\beta}^{k-1}\|_0 - 1$.

**Lemma 3.10** (Backward Inverse). *Suppose* $\mathbf{B}^{-1}$ *is known,*

*where* $\mathbf{B}^{-1} = \begin{pmatrix} \mathbf{G}_{(n-1)\times(n-1)} & \mathbf{w} \\ \mathbf{z}^T & \gamma \end{pmatrix}_{n \times n}$, $\mathbf{B} = \begin{pmatrix} \mathbf{A}_{(n-1)\times(n-1)} & \mathbf{u} \\ \mathbf{v}^T & \alpha \end{pmatrix}_{n \times n}$. *Then the inverse of* $\mathbf{A}$ *could be updated as:*

$$\mathbf{A}^{-1} = \mathbf{G} - \mathbf{w}\mathbf{z}^T/\gamma. \qquad (9)$$

Lemma 3.10 is a straightforward corollary of Lemma 3.2. We can derive the following optimal feature elimination criterion by applying Lemma 3.10.

**Theorem 3.11.** *Let* $\mathbf{C}_{k-1} = \left(\mathbf{X}_{S_{k-1}}^T \mathbf{X}_{S_{k-1}}\right)^{-1}$, $\mathbf{e}_j = (\delta_{1i}, \delta_{2i}, \cdots, \delta_{ii}, \cdots, \delta_{|S_{k-1}|i})^T \in \mathbb{R}^{|S_{k-1}|}$, *where* $j$ *represents the* $i$-th *element of* $S_{k-1}$ *for* $i = 1, 2, \ldots, |S_{k-1}|$. *The Kronecker delta function* $\delta_{ab}$ *is defined as* $\delta_{ab} = 1$ *if* $a = b$, *and* $\delta_{ab} = 0$ *otherwise. Then, problem* (Q2) *is equivalent (in the sense of identifying the true subset) to*

$$\operatorname*{argmax}_{j \in S_{k-1}} \quad \mathbf{y}^T \mathbf{X}_{S_{k-1}} \left(\mathbf{I} - \mathbf{e}_j \mathbf{e}_j^T\right) \left(\mathbf{C}_{k-1} - \frac{\mathbf{C}_{k-1}\mathbf{e}_j \mathbf{e}_j^T \mathbf{C}_{k-1}}{\mathbf{e}_j^T \mathbf{C}_{k-1} \mathbf{e}_j}\right) \left(\mathbf{I} - \mathbf{e}_j \mathbf{e}_j^T\right) \mathbf{X}_{S_{k-1}}^T \mathbf{y}. \quad (10)$$

The proof of Theorem 3.11 is provided in Appendix C.

**Definition 3.12** (**Objective-based Elimination**). By Theorem 3.11, the new criterion for feature importance inside the support set could be formulated as criterion (10).

*Remark* 3.13. When $\mathbf{X}_j$ and $\mathbf{X}_{S_{k-1}\setminus\{j\}}$ are orthogonal, the objective-based elimination criterion (10) degenerates into the Wald-T based criterion (4). A detailed proof is provided in Appendix D.

*Remark* 3.14 (Comprehensive Combination Effect). The new criterion (10) takes into account the impact on the other features in $S_{k-1}$ resulting from the elimination of feature $j$.

*Remark* 3.15 (Computational Efficiency). The matrix $\mathbf{C}_{k-1}$ or Cholesky decomposition of $\mathbf{X}_{S_{k-1}}^T \mathbf{X}_{S_{k-1}}$ has been already computed in the previous step when updating the

coefficients on the support set $S_{k-1}$, so the inversion term in criterion (10) does not incur additional magnitude of computational cost. See Appendix O for comparison in runtime.

### 3.3. Enhanced Algorithms for Best Subset Selection

As mentioned in the Introduction, heuristic subset selection algorithms generally determine selection and elimination based on (3) and (4), with different algorithms arising from various combination strategies of these criteria. By leveraging the optimal criteria in Section 3.2, we can perform Meta-Substitution of the objective-based criteria (8) and (10) into classical algorithms like MP, OMP, CoSaMP, IHT, and (A)BESS, resulting in an enhanced algorithm family, as shown in Figure 2.

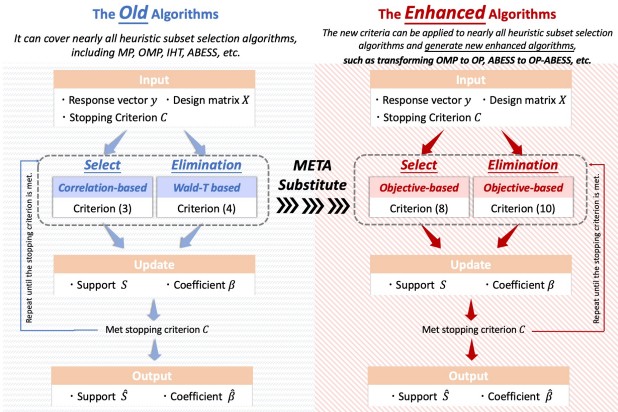

*Figure 2.* Left: The original algorithmic workflow based on heuristic subset selection criteria. Right: By replacing the classic criteria with optimal objective-based criteria, the original heuristic algorithms can be updated to yield new algorithms accordingly.

We classify subset selection algorithms into three categories based on their combination strategies for feature selection and elimination, providing one representative for each to show how Meta-Substitution generates new algorithms:

**Select-Only** : This type of algorithm greedily selects feature at each step. Examples include MP, OMP, etc. For instance, OMP can be upgraded to Optimal Pursuit (OP) algorithm, as described in Algorithm 1 in Appendix E.

**Select-First, Eliminate-Next**: This type of algorithms first selects the features and then removes the irrelevant ones. Examples include CoSaMP, IHT, HTP, etc. For instance, CoSaMP can be enhanced to CoSaOP, as detailed in Algorithm 2 in Appendix E.

**Exchange-Based**: This class of algorithms performs subset selection by swapping irrelevant features in the active set with significant features outside the active set. Notably, (A)BESS (Zhu et al., 2020), currently serving as a benchmark method in the subset selection field, belongs to this category. It can be upgraded to the OP-(A)BESS algorithm, as outlined in Algorithm 3 and 4 in Appendix E.

Beyond these examples, other greedy subset selection algorithms can also be enhanced through meta substitution scheme. We will demonstrate that these enhanced algorithms not only retain the original theoretical properties but also achieve significant meta-gains across various tasks, scenarios and evaluation metrics.

# 4. Theory of Optimal Subset Selection Criteria

In this section, we establish theory of optimal subset selection criteria (8) and (10). Define the function $f(S)$ as:

$$f(S) \triangleq \min_{\boldsymbol{\beta}} \|\mathbf{y} - \mathbf{X}\boldsymbol{\beta}\|_2^2$$
$$\text{s. t. } \text{supp}(\boldsymbol{\beta}) = S.$$

With this definition, we present the following theorems.

**Theorem 4.1.** *For index $j^*$ selected by criterion (8),*

$$f\left(S \cup \{j^*\}\right) \leq f\left(S \cup \{j\}\right), \ \forall j \in S^c.$$

**Theorem 4.2.** *For index $j^*$ selected by criterion (10),*

$$f\left(S \backslash \{j^*\}\right) \leq f\left(S \backslash \{j\}\right), \ \forall j \in S.$$

Theorems 4.1 and 4.2 summarize the previous discussion, demonstrating that criteria (8) and (10) serve as the optimal decisions in the current subset selection process.

**Theorem 4.3.** *The computational complexities of OMP and OP, CoSaMP and CoSaOP, as well as (A)BESS and OP-(A)BESS, are of the same order of magnitude.*

Theorem 4.3 follows from Remark 3.7 and Remark 3.15, and it indicates that the enhanced algorithms have the same computational complexity as the original algorithms. Now, we take the CoSaOP algorithm (Select-First, Eliminate-Next) as an example to illustrate how the enhanced algorithms retain the theoretical properties of the original algorithm. The subsequent Lemmas 4.4–4.8 and Theorem 4.9 correspond directly to theoretical results of CoSaMP in (Needell & Tropp, 2009).

The basic assumption and proofs of Lemmas 4.4–4.8 are provided in Appendices F-J. And the proof of convergence property of CoSaOP (Theorem 4.9) is presented in Appendix K.

**Lemma 4.4** (Identification). *The set $\Omega$ selected by CoSaOP during identification stage (Step 6 in Algorithm 2) at iteration $k$ satisfies*

$$\left\|\left(\boldsymbol{\beta}^{k-1} - \boldsymbol{\beta}^*\right)|_{\Omega^c}\right\|_2 \leq 0.2353 \left\|\boldsymbol{\beta}^{k-1} - \boldsymbol{\beta}^*\right\|_2 + 2.4 \left\|\boldsymbol{\epsilon}\right\|_2.$$

**Lemma 4.5** (Support merger). *Let $\Omega$ be a set containing at most $2K$ indices. Then, the set $U = \Omega \cup supp(\boldsymbol{\beta}^{k-1})$ contains at most $3K$ indices, and it holds that*

$$\left\|\boldsymbol{\beta}^*|_{U^c}\right\|_2 \leq \left\|\left(\boldsymbol{\beta}^* - \boldsymbol{\beta}^{k-1}\right)|_{\Omega^c}\right\|_2.$$

**Lemma 4.6** (Estimation). *Let $U$ be a set containing at most $3K$ indices. The least-squares estimate $\mathbf{a}$ is formulated as*

$$\mathbf{a}|_U = \left(\mathbf{X}_U^T \mathbf{X}_U\right)^{-1} \mathbf{X}_U^T \mathbf{y}, \quad \mathbf{a}|_{U^c} = \mathbf{0}.$$

*Then, the following bound holds:*

$$\|\boldsymbol{\beta}^* - \mathbf{a}\|_2 \leq 1.112 \|\boldsymbol{\beta}^*|_{U^c}\|_2 + 1.06\|\boldsymbol{\epsilon}\|_2.$$

**Lemma 4.7** (Elimination). *Let $S_k$ be the index set selected by CoSaOP from $U$ according to criterion (10). Then,*

$$\|\boldsymbol{\beta}^* - \tilde{\mathbf{a}}_{S_k}\|_2 \leq (2 + \delta) \|\boldsymbol{\beta}^* - \mathbf{a}\|_2,$$

*where $\tilde{\mathbf{a}}_{S_k,j} = \mathbf{a}_j$ for $j \in S_k$, $\tilde{\mathbf{a}}_{S_k,j} = 0$ for $j \in S_k^c$, and $0 \leq \delta \leq 1$ is a constant related to the orthogonality of $\mathbf{X}_U$.*

**Lemma 4.8** (Least square estimation). *Let $\boldsymbol{\beta}^k$ denote the least-square estimation on $S_k$ obtained by CoSaOP. We have*

$$\|\boldsymbol{\beta}^k - \boldsymbol{\beta}^*\|_2 \leq 1.106 \|\boldsymbol{\beta}^* - \tilde{\mathbf{a}}_{S_k}\|_2 + 2.109\|\boldsymbol{\epsilon}\|_2.$$

Therefore, based on Lemmas 4.4-4.8, we arrive at the following theorem.

**Theorem 4.9** (CoSaOP). *When the approximation error is large in comparison with the noise, the CoSaOP algorithm achieves a linear convergence rate, expressed as*

$$\|\boldsymbol{\beta}^k - \boldsymbol{\beta}^*\|_2 \leq 0.869\|\boldsymbol{\beta}^{k-1} - \boldsymbol{\beta}^*\|_2 + 14.482\|\boldsymbol{\epsilon}\|_2.$$

In real-world scenarios, features are often highly correlated, making the RIP assumption, commonly used in classical best subset selection algorithms, invalid. Theorems 4.10 and 4.11 further demonstrate the significant advantages of criteria (8) and (10) over classical criteria (3) and (4) in the presence of high feature correlation. For the proofs and further explanations, please refer to the Appendices L-N.

**Theorem 4.10.** *Suppose the true subset $S^*$ contains indices $(i, j)$, where the correlation between feature $\mathbf{X}_i$ and $\mathbf{X}_j$ is $\rho = \frac{|\mathbf{X}_i^T \mathbf{X}_j|}{\|\mathbf{X}_i\|_2\|\mathbf{X}_j\|_2}$. Assuming the current support set $S$ already includes feature $i$, then the classical correlation-based criterion (3) for feature $j$ satisfies:*

$$\frac{|\mathbf{r}^{k^T}\mathbf{X}_j|}{\|\mathbf{X}_j\|_2} \leq \sqrt{1 - \rho^2}\|\mathbf{r}^k\|_2, \tag{11}$$

*while the objective-based criterion (8) satisfies*

$$\frac{\left(\mathbf{r}^{k^T}\mathbf{X}_j\right)^2}{\mathbf{X}_j^T \left(\mathbf{I} - \mathbf{X}_S \left(\mathbf{X}_S^T\mathbf{X}_S\right)^{-1} \mathbf{X}_S^T\right) \mathbf{X}_j} \geq \frac{1}{1 - \rho^2} \left(\frac{\mathbf{r}^{k^T}\mathbf{X}_j}{\|\mathbf{X}_j\|_2}\right)^2. \tag{12}$$

**Theorem 4.11.** *(1) The upper bound of the objective-based criterion (10) is $\|\mathbf{y}\|_2^2$. If the true subset $S^*$ is contained within the current subset $S$, then for all $j_m \in S \setminus S^*$,*

$$\|\mathbf{y}\|_2^2 - \|\boldsymbol{\epsilon}\|_2^2 \leq (criterion (10) for j_m) \leq \|\mathbf{y}\|_2^2.$$

*And in noiseless scenario,*

$$j_m \in \operatorname{argmax}_{j \in S} \; objective\text{-}based \; criterion \; (10).$$

*(2) Suppose a feature $\mathbf{X}_p$ in the current subset is pesudo-correlated with an important feature $\mathbf{X}_i$ in the true sub-set (with correlation $1 - \mu$). When $\mu$ is sufficiently small, classical T-statistics based criteria (4) could erroneously discard true features even in simple cases like $S = S^* \cup \{p\}$, whereas proposed criterion (10) could correctly identify and remove the spurious feature $\mathbf{X}_p$.*

We also evaluate the algorithm's performance under high feature correlations to validate the theories through their marked improvements. See Appendix P for details.

## 5. Experiments

We conducted experiments on two typical subset selection problems: compressed sensing and sparse regression, to demonstrate the meta-gains achieved by the new algorithms developed through meta-substitution. The comparison involves representative algorithms[1] from the three categories introduced in Section 3.3: OP, CoSaOP, and OP-(A)BESS, evaluated against classical subset selection methods: OMP, CoSaMP, and (A)BESS. Their performance is evaluated across multiple metrics, including the number of successful recoveries, NMSE, $R^2$, and runtime, highlighting the superiority of the enhanced algorithms from various perspectives.

### 5.1. Compressed Sensing

In this task, there is a ground truth signal for $\boldsymbol{\beta}$. The goal is to recover the high-dimensional sparse vector $\boldsymbol{\beta} \in \mathbb{R}^{p \times 1}$, given the noisy low-dimensional observation $\mathbf{y} \in \mathbb{R}^{n \times 1}$ and design matrix $\mathbf{X} \in \mathbb{R}^{n \times p}$ ($n \ll p$).

#### 5.1.1. SYNTHETIC SPARSE DATA

In this experiment, we randomly generate $\boldsymbol{\beta}$ with a dimensionality of $p = 200$ and a sparsity level of $K = 10$. The design matrix $\mathbf{X}$ is a $n \times p$ random Gaussian matrix. Let $S^*$ represent the true support set of the signal and $\hat{S}$ the estimated support set, recovery is deemed successful if $\hat{S} = S^*$. For each algorithm, we conduct 500 independent runs and record the number of successful recoveries as shown in Figure 3. The first row illustrates the variation in the number of successful recoveries for three groups of algorithms as the sampling rate increases from 25% to 50% under a fixed SNR of 15. The second row, in contrast, fixes the sampling rate at 25% and shows how the number of successful recoveries changes as the SNR increases from 15 to 25.

As a result, in the most extreme compressed scenario with a measurement rate of 0.25, OP achieves nearly $\mathbf{3\times}$ improvement over OMP. CoSaOP consistently outperforms

---

[1]Matlab codes are available at https://github.com/ZhihanZhu-math/Optimal_Pursuit_public.

CoSaMP by at least $\mathbf{4\times}$ in all scenarios, with a maximum improvement of nearly $\mathbf{7\times}$ under SNR = 21, as shown in subplot (e). While (A)BESS, as the state-of-the-art algorithm for this task, demonstrates great performance across various settings, OP-(A)BESS still manages to achieve observable meta-gains beyond it. Even in near-saturation scenarios where (A)BESS achieves almost complete recovery, the meta-gain remains evident. Thus, the proposed algorithms (blue bars) consistently outperform their classical counterparts (green bars) across all experimental conditions, delivering significant improvements in recovery rates.

We also compare the computational time in Appendix O. In Appendix P, we conduct further experiment under high feature correlations with small sample sizes, high-dimensional features, and high noise levels. The experimental results further highlight the signiaficant advantages of the Optimal Pursuit-enhanced algorithms in extreme scenarios.

#### 5.1.2. SIGNALS FROM *AudioSet*

After transforming audio signals into the DCT domain, they exhibit transform sparsity (Donoho, 2006). Therefore, we test our method using real-world audio data. The data presented here is randomly sampled from the *AudioSet* dataset (Gemmeke et al., 2017), consisting of 4 audio signals with full dimensionality $p = 480$. These signals have approximately 20–40 non-zero entries ($K$) in the DCT domain, with the number of observations fixed at $n = 150$. The normalized mean squared error (NMSE), defined as NMSE $= \|\hat{\boldsymbol{\beta}} - \boldsymbol{\beta}^*\|_2^2 / \|\boldsymbol{\beta}^*\|_2^2$, is used to quantify the recovery performance. We conducted 100 random experiments, and the results are summarized in Table 1.

The proposed algorithms consistently exhibit significant performance improvements. In particular, OP shows observable meta-gain compared to OMP. Although CoSaMP algorithm already performs well on this task (with small NMSE and std), CoSaOP still achieves **near-order-of-magnitude** improvement. Similarly, OP-(A)BESS achieves near-order-of-magnitude improvements **in both mean and std of NMSE** in multiple trials, indicating that the algorithm's accuracy and stability have been significantly enhanced.

### 5.2. Sparse Regression

In sparse regression tasks, $\boldsymbol{\beta}$ does not have a ground truth. The goal is to select sparse features that provide a better explanation of target variable $\mathbf{y}$. Therefore, similar to the metric used in sparse regression evaluations in (Qian et al., 2017; Das & Kempe, 2018), we quantify the explanatory ability of the features using Coefficient of Determination: $R^2 = 1 - \sum_{i=1}^{n}(y_i - \hat{y}_i)^2 / \sum_{i=1}^{n}(y_i - \bar{\mathbf{y}})^2$.

We utilize six real-world datasets in our experiments: (1) Boston Housing Data (Pedregosa et al., 2011), (2) California Housing Data (Pedregosa et al., 2011), (3) Superconduc-

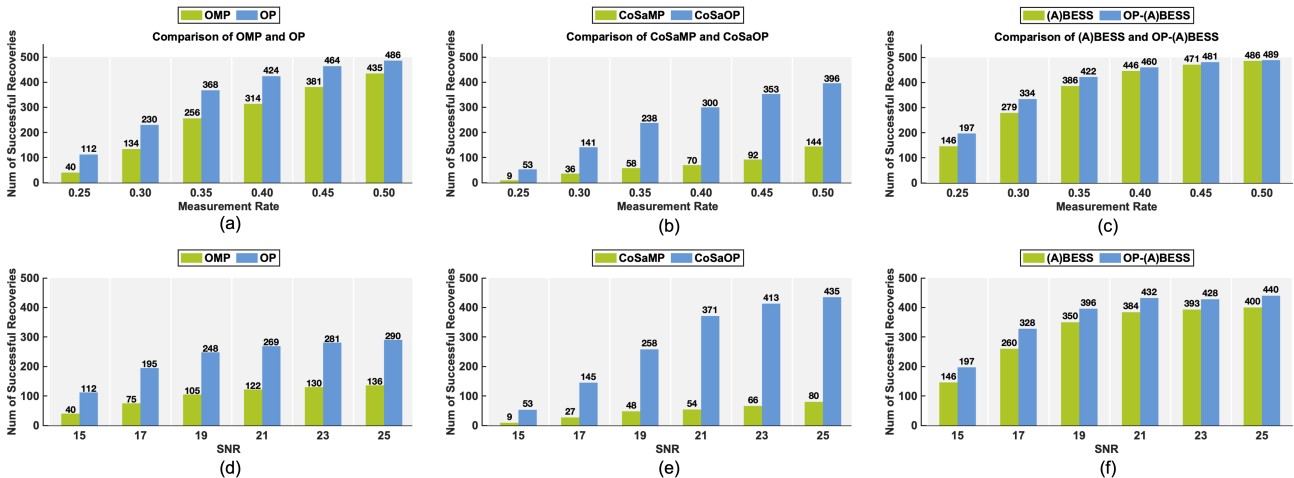

*Figure 3.* Meta-gain comparison of three kinds of subset selection algorithms. Row one: different sampling rates (SNR = 15). Row two: Varying SNRs (measurement rate = 0.25).

*Table 1.* Reconstruction NMSE (mean $\pm$ std) for Signals from *AudioSet*. Meta-gains are highlighted in **red**, with the best in **bold** and the second-best underlined. (Audio 1-4: -0SdAVK79lg.wav, _qxgIqI0uA.wav, _0bN5mYLXb0.wav, _0Jd6JJeyJ4.wav)

| | NMSE (OMP & OP) | | | NMSE (CoSaMP & CoSaOP) | | | NMSE ((A)BESS & OP-(A)BESS) | | |
|---|---|---|---|---|---|---|---|---|---|
| Audio Set | OMP | OP | Gains | CoSaMP | CoSaOP | Gains | (A)BESS | OP-(A)BESS | Gains |
| Audio 1 | 9.45E-04 (1.35E-03) | 7.69E-04 (5.78E-04) | **19%** | 2.36E-03 (1.22E-03) | 1.64E-03 (1.36E-03) | **31%** | 1.41E-03 (7.89E-03) | **6.22E-04 (2.64E-04)** | **56%** |
| Audio 2 | 5.79E-03 (7.89E-03) | 5.49E-03 (8.87E-03) | **5%** | 1.87E-03 (6.05E-04) | **6.36E-04 (2.01E-04)** | **66%** | 6.71E-03 (6.07E-02) | **6.36E-04 (2.01E-04)** | **91%** |
| Audio 3 | 1.46E-03 (3.13E-03) | 1.18E-03 (2.14E-03) | **19%** | 1.54E-03 (5.86E-04) | 5.52E-04 (2.03E-04) | **64%** | 2.84E-03 (2.29E-02) | 5.52E-04 (2.03E-04) | **81%** |
| Audio 4 | 4.05E-02 (1.03E-01) | 3.55E-02 (9.70E-02) | **12%** | 2.85E-03 (9.84E-04) | **8.22E-04 (2.22E-04)** | **71%** | 2.65E-02 (1.07E-01) | 1.86E-02 (1.12E-01) | **30%** |

tivity Data (Hamidieh, 2018), (4) House 16H (Vanschoren, 2014), (5) Prostate.v8.egen (Lin & Pan, 2024; Hastie et al., 2017), and (6) Spectra (The MathWorks, Inc., 2025). For the first five datasets, which have a relatively small number of features, we augment the feature space by constructing polynomial features. The dimensions of the six datasets are as follows: $n_1 = 506$, $p_1 = 104$; $n_2 = 500$ (randomly sampled), $p_2 = 164$; $n_3 = 500$ (randomly sampled), $p_3 = 80$; $n_4 = 500$ (randomly sampled), $p_4 = 153$; $n_5 = 500$ (randomly sampled), $p_5 = 253$; and $n_6 = 60$, $p_6 = 401$.

The experimental results in Figure 4 demonstrate the enhanced algorithms driven by new criteria consistently yield significant meta-gains across different datasets and varying numbers of selected features $K$. The enhanced algorithms outperform the original algorithms, achieving gains equivalent to selecting 10 more features. Notably, OP and OP-(A)BESS demonstrate approximately a 0.1 improvement in $R^2$. And even though CoSaMP fails in this regression task (see Appendix Q for detailed reasons), the enhanced CoSaOP remains effective and delivers strong performance.

## 6. Discussion

In this section, we discuss the potential applications of the optimal pursuit idea in a broader range of machine learning tasks and scenarios.

### 6.1. Best Subset Selection with Ultra-high Dimensions: Optimal Gradient Pursuit

As noted in Remark 3.7 and Remark 3.15, the algorithmic complexity of the original methods and our optimal pursuit-enhanced algorithm remains at the same order of magnitude, since both rely on solving the least squares problem over a given subset $S$, which involves a linear system solved via Cholesky decomposition. However, in ultra-high-dimensional settings, solving the least squares problem over a given subset, i.e., solving a linear system, can be computationally prohibitive, affecting both the original and enhanced algorithms. In fact, updating coefficients via least squares within a given subset is equivalent to using the Newton method for updates. (Blumensath & Davies, 2008) proposed Gradient Pursuit, which follows the same correlation-based selection strategy in the basis selection step. However, in the coefficient update step, instead of Newton's method, it employs gradient-based updates within a given support set, significantly reducing computational overhead in ultra-high-dimensional scenarios.

Here, we want to emphasize that our proposed optimal pursuit idea can also be applied to Gradient Pursuit, which we refer to as Optimal Gradient Pursuit. Optimal Gradient Pursuit simultaneously considers both support set updates (fea-

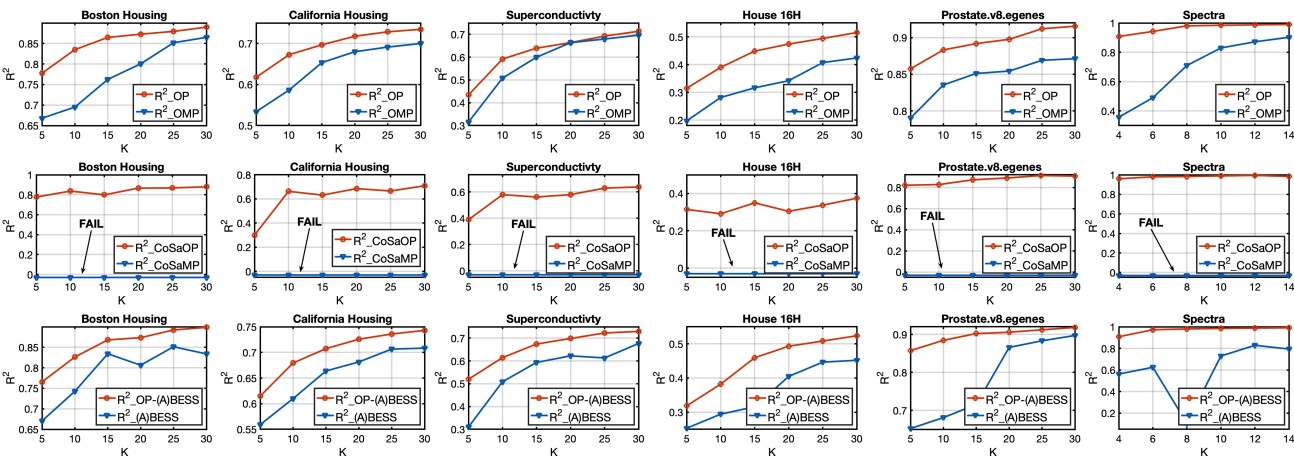

*Figure 4.* Rows 1–3 present the meta-gains in feature representation capability ($R^2$, closer to 1 is better) for the Boston Housing, California Housing, Superconductivity datasets, House 16H, Prostate.v8.egenes, and Spectra datasets, respectively, across three algorithms as the number of selected features $K$ varies. See Appendix R for cross validation performance in prediction.

ture individual significance) and coefficient updates (feature interaction), while maintaining the same order of computational complexity as Gradient Pursuit.

For the derivation of the Optimal Gradient Pursuit criteria, the proof of its theoretical properties and the numerical experiments, please refer to the Appendices S-U. Optimal Gradient Pursuit serves as an acceleration scheme for Optimal Pursuit, which is applicable to ultra-high dimensional settings, and the gradient-based updates is practical for best subset selection problems with general objective functions.

### 6.2. Unsupervised Learning: Column Subset Selection

Column Subset Selection (CSS) and PCA are both important dimensionality reduction methods with widespread applications in unsupervised learning (Belhadji et al., 2020). The goal of CSS is to select a subset of important columns (features) from a dataset that can better represent the entire dataset, formulated as:

$$\min_{S,\mathbf{B}} \|\mathbf{X} - \mathbf{X}_S \mathbf{B}\|_F^2$$
$$\text{s. t. } |S| \leq K.$$

In fact, this problem can also be viewed as a special case of best subset selection problem. We have extended the optimal pursuit criterion to the CSS task, demonstrating the advantages of our proposed criteria over classical criteria in this setting. For the derivation of the criteria and numerical experiments, please refer to the Appendix V.

### 6.3. Complex Signal Processing

Although our paper primarily discusses these theories and methods in the real domain, they can be directly extended to the complex domain. A classic example in complex signal processing is line spectrum estimation. This problem can be viewed as a special case of feature

selection, where the features are continuous in the frequency domain and exhibit a specific structure: $\mathbf{v}(f) = \left(1, e^{-j2\pi f}, e^{-j2\pi 2f}, \ldots, e^{-j2\pi(n-1)f}\right)^T$. The goal is to decompose a given complex signal into its frequency components, a problem widely applied in modern wireless communications (Mamandipoor et al., 2016). For the numerical experiments of Optimal Pursuit in complex signal processing, please refer to the Appendix W.

## 7. Conclusion

This paper proposed two novel objective-based criteria for feature selection and elimination in best subset selection. By revisiting classical criteria in traditional algorithms through the lens of block coordinate descent, we revealed that they only reflect a one-step variation of the objective function. Building on this, we formulated exact optimization subproblems for feature selection and elimination, deriving explicit solutions using forward and backward matrix inversion. The proposed criteria account for both individual feature significance and interactions, proving optimal for subset selection. Replacing classical criteria with the proposed ones, we developed enhanced algorithms that retain the original theoretical guarantees while achieving notable performance gains across various tasks, such as compressed sensing and sparse regression, all without added computational cost.

The results affirm the advantages of the new criteria both theoretically and practically, opening new avenues for improving best subset selection algorithms. Future work may consider: (1) integrating the proposed criteria into arbitrary greedy subset selection algorithms to develop enhanced methods and application on structured sparse learning (Huang et al., 2009), (2) developing optimal selection and elimination criteria for general objective functions, (3) investigating statistical inference theories of the new criteria.

## Acknowledgements

This research was supported by National Key R&D Program of China under grant 2021YFA1003303.

The authors would like to thank all the reviewers for their helpful comments. Their suggestions have helped us enhance our theories and experiments to present a more comprehensive demonstration of the effectiveness of our work.

## Impact Statement

This paper presents work whose goal is to advance the field of Machine Learning. There are many potential societal consequences of our work, none which we feel must be specifically highlighted here.

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

## A. Proof of Lemma 3.1

*Proof.* Let $S = \text{supp}(\boldsymbol{\beta})$. Since the solution $\hat{\boldsymbol{\beta}}$ of (P1) is obtained via least squares on $S$, i.e.,

$$\boldsymbol{\beta}_S = \left(\mathbf{X}_S^T \mathbf{X}_S\right)^{-1} \mathbf{X}_S^T \mathbf{y}, \quad \boldsymbol{\beta}_{S^c} = 0. \tag{13}$$

Then, substituting (13) into (P1) directly yields (P2). $\square$

## B. Proof of Theorem 3.3

*Proof.* Denoting $S_{k-1} = \text{supp}(\boldsymbol{\beta}^{k-1})$ and $S = \text{supp}(\boldsymbol{\beta})$, we can rewrite (P2) as follows:

$$\underset{\boldsymbol{\beta}}{\text{argmax}} \ \mathbf{y}^T \mathbf{X}_S \left(\mathbf{X}_S^T \mathbf{X}_S\right)^{-1} \mathbf{X}_S^T \mathbf{y} \tag{14}$$

$$\text{s.t.} \ \ S_{k-1} \subset S, \ \ |S| = |S_{k-1}| + 1,$$

where the inverse term $\left(\mathbf{X}_S^T \mathbf{X}_S\right)^{-1}$ can be expressed as

$$\left(\mathbf{X}_S^T \mathbf{X}_S\right)^{-1} = \begin{pmatrix} \mathbf{X}_{S_{k-1}}^T \\ \mathbf{X}_j^T \end{pmatrix} \begin{pmatrix} \mathbf{X}_{S_{k-1}} & \mathbf{X}_j \end{pmatrix} = \begin{pmatrix} \mathbf{X}_{S_{k-1}}^T \mathbf{X}_{S_{k-1}} & \mathbf{X}_{S_{k-1}}^T \mathbf{X}_j \\ \mathbf{X}_j^T \mathbf{X}_{S_{k-1}} & \mathbf{X}_j^T \mathbf{X}_j \end{pmatrix}.$$

According to Lemma 3.2,

$$\left(\mathbf{X}_S^T \mathbf{X}_S\right)^{-1} = \begin{pmatrix} \mathbf{A}^{-1} + \mathbf{A}^{-1}\mathbf{u}t^{-1}\mathbf{v}^T\mathbf{A}^{-1} & -\mathbf{A}^{-1}\mathbf{u}t^{-1} \\ -t^{-1}\mathbf{v}^T\mathbf{A}^{-1} & t^{-1} \end{pmatrix},$$

where $\mathbf{u} = \mathbf{X}_{S_{k-1}}^T \mathbf{X}_j$, $\mathbf{v}^T = \mathbf{X}_j^T \mathbf{X}_{S_{k-1}}$, and $\mathbf{A}^{-1} = \left(\mathbf{X}_{S_{k-1}}^T \mathbf{X}_{S_{k-1}}\right)^{-1}$, which was computed in step $k-1$ and is already known. Additionally, $t$ is given by

$$t = \mathbf{X}_j^T \mathbf{X}_j - \mathbf{v}^T \mathbf{A}^{-1} \mathbf{u} = \mathbf{X}_j^T \mathbf{X}_j - \mathbf{X}_j^T \mathbf{X}_{S_{k-1}} \left(\mathbf{X}_{S_{k-1}}^T \mathbf{X}_{S_{k-1}}\right)^{-1} \mathbf{X}_{S_{k-1}}^T \mathbf{X}_j.$$

Then, $\mathbf{X}_S \left(\mathbf{X}_S^T \mathbf{X}_S\right)^{-1} \mathbf{X}_S^T$ can be reformulated as

$$\mathbf{X}_S \left(\mathbf{X}_S^T \mathbf{X}_S\right)^{-1} \mathbf{X}_S^T = \begin{pmatrix} \mathbf{X}_{S_{k-1}} & \mathbf{X}_j \end{pmatrix} \begin{pmatrix} \mathbf{A}^{-1} + \mathbf{A}^{-1}\mathbf{u}t^{-1}\mathbf{v}^T\mathbf{A}^{-1} & -\mathbf{A}^{-1}\mathbf{u}t^{-1} \\ -t^{-1}\mathbf{v}^T\mathbf{A}^{-1} & t^{-1} \end{pmatrix} \begin{pmatrix} \mathbf{X}_{S_{k-1}}^T \\ \mathbf{X}_j^T \end{pmatrix}$$

$$= \mathbf{X}_{S_{k-1}} \left(\mathbf{A}^{-1} + \mathbf{A}^{-1}\mathbf{u}t^{-1}\mathbf{v}^T\mathbf{A}^{-1}\right) \mathbf{X}_{S_{k-1}}^T - \mathbf{X}_{S_{k-1}}\mathbf{A}^{-1}\mathbf{u}t^{-1}\mathbf{X}_j^T - \mathbf{X}_j t^{-1}\mathbf{v}^T\mathbf{A}^{-1}\mathbf{X}_{S_{k-1}}^T + t^{-1}\mathbf{X}_j\mathbf{X}_j^T$$

$$= \mathbf{X}_{S_{k-1}} \left(\mathbf{X}_{S_{k-1}}^T \mathbf{X}_{S_{k-1}}\right)^{-1} \mathbf{X}_{S_{k-1}} + \frac{\mathbf{X}_{S_{k-1}} \left(\mathbf{X}_{S_{k-1}}^T \mathbf{X}_{S_{k-1}}\right)^{-1} \mathbf{X}_{S_{k-1}}^T \mathbf{X}_j \mathbf{X}_j^T \mathbf{X}_{S_{k-1}} \left(\mathbf{X}_{S_{k-1}}^T \mathbf{X}_{S_{k-1}}\right)^{-1} \mathbf{X}_{S_{k-1}}^T}{\mathbf{X}_j^T \mathbf{X}_j - \mathbf{X}_j^T \mathbf{X}_{S_{k-1}} \left(\mathbf{X}_{S_{k-1}}^T \mathbf{X}_{S_{k-1}}\right)^{-1} \mathbf{X}_{S_{k-1}}^T \mathbf{X}_j}$$

$$- \frac{\mathbf{X}_{S_{k-1}} \left(\mathbf{X}_{S_{k-1}}^T \mathbf{X}_{S_{k-1}}\right)^{-1} \mathbf{X}_{S_{k-1}}^T \mathbf{X}_j \mathbf{X}_j^T + \mathbf{X}_j \mathbf{X}_j^T \mathbf{X}_{S_{k-1}} \left(\mathbf{X}_{S_{k-1}}^T \mathbf{X}_{S_{k-1}}\right)^{-1} \mathbf{X}_{S_{k-1}}^T - \mathbf{X}_j \mathbf{X}_j^T}{\mathbf{X}_j^T \mathbf{X}_j - \mathbf{X}_j^T \mathbf{X}_{S_{k-1}} \left(\mathbf{X}_{S_{k-1}}^T \mathbf{X}_{S_{k-1}}\right)^{-1} \mathbf{X}_{S_{k-1}}^T \mathbf{X}_j}$$

$$= \mathbf{X}_{S_{k-1}} \left(\mathbf{X}_{S_{k-1}}^T \mathbf{X}_{S_{k-1}}\right)^{-1} \mathbf{X}_{S_{k-1}}$$

$$+ \frac{\left(\mathbf{I} - \mathbf{X}_{S_{k-1}} \left(\mathbf{X}_{S_{k-1}}^T \mathbf{X}_{S_{k-1}}\right)^{-1} \mathbf{X}_{S_{k-1}}^T\right) \mathbf{X}_j \mathbf{X}_j^T \left(\mathbf{I} - \mathbf{X}_{S_{k-1}} \left(\mathbf{X}_{S_{k-1}}^T \mathbf{X}_{S_{k-1}}\right)^{-1} \mathbf{X}_{S_{k-1}}^T\right)}{\mathbf{X}_j^T \left(\mathbf{I} - \mathbf{X}_{S_{k-1}} \left(\mathbf{X}_{S_{k-1}}^T \mathbf{X}_{S_{k-1}}\right)^{-1} \mathbf{X}_{S_{k-1}}^T\right) \mathbf{X}_j}.$$

Therefore, problem (P2) is equivalent to

$$
\underset{j \in S_{k-1}^c}{\mathrm{argmax}} \; \frac{\mathbf{y}^T \left( \mathbf{I} - \mathbf{X}_{S_{k-1}} \left( \mathbf{X}_{S_{k-1}}^T \mathbf{X}_{S_{k-1}} \right)^{-1} \mathbf{X}_{S_{k-1}}^T \right) \mathbf{X}_j \mathbf{X}_j^T \left( \mathbf{I} - \mathbf{X}_{S_{k-1}} \left( \mathbf{X}_{S_{k-1}}^T \mathbf{X}_{S_{k-1}} \right)^{-1} \mathbf{X}_{S_{k-1}}^T \right) \mathbf{y}}{\mathbf{X}_j^T \left( \mathbf{I} - \mathbf{X}_{S_{k-1}} \left( \mathbf{X}_{S_{k-1}}^T \mathbf{X}_{S_{k-1}} \right)^{-1} \mathbf{X}_{S_{k-1}}^T \right) \mathbf{X}_j}
$$

$$
= \underset{j \in S_{k-1}^c}{\mathrm{argmax}} \; \frac{\left( \mathbf{r}^{k^T} \mathbf{X}_j \right)^2}{\mathbf{X}_j^T \left( \mathbf{I} - \mathbf{X}_{S_{k-1}} \left( \mathbf{X}_{S_{k-1}}^T \mathbf{X}_{S_{k-1}} \right)^{-1} \mathbf{X}_{S_{k-1}}^T \right) \mathbf{X}_j}, \tag{15}
$$

where $\mathbf{r}^k = \mathbf{y} - \mathbf{X}_{S_{k-1}} \boldsymbol{\beta}^{k-1}$ represents the residual obtained from the $(k-1)$-th update, and $\boldsymbol{\beta}^{k-1} = \left( \mathbf{X}_{S_{k-1}}^T \mathbf{X}_{S_{k-1}} \right)^{-1} \mathbf{X}_{S_{k-1}}^T \mathbf{y}$ is the coefficient estimate from step $k-1$. $\qquad \square$

## C. Proof of Theorem 3.11

*Proof.* Denoting $S_{k-1} = \mathrm{supp}(\boldsymbol{\beta}^{k-1})$ and $S = \mathrm{supp}(\boldsymbol{\beta})$, we can rewrite (Q2) as follows:

$$
\underset{\boldsymbol{\beta}}{\mathrm{argmax}} \; \mathbf{y}^T \mathbf{X}_S \left( \mathbf{X}_S^T \mathbf{X}_S \right)^{-1} \mathbf{X}_S^T \mathbf{y} \tag{16}
$$

$$
\mathrm{s.\,t.} \; \; S \subset S_{k-1}, \; \; |S| = |S_{k-1}| - 1,
$$

where the inverse term $\left( \mathbf{X}_S^T \mathbf{X}_S \right)^{-1}$ is already known, as it was computed in step $k-1$.

Let $\mathbf{e}_i = (\delta_{1i}, \delta_{2i}, \cdots, \delta_{ii}, \cdots, \delta_{|S_{k-1}|i})^T \in \mathbb{R}^{|S_{k-1}|}$, for all $i = 1, 2, \cdots, |S_{k-1}|$, where the Kronecker delta function $\delta_{ij}$ is defined as

$$
\delta_{ij} = \begin{cases} 1, & \text{if } i = j, \\ 0, & \text{otherwise.} \end{cases}
$$

Let $\mathbf{P}_i = \left( \mathbf{e}_1, \cdots, \mathbf{e}_{i-1}, \mathbf{e}_{i+1}, \cdots, \mathbf{e}_{|S_{k-1}|} \right) \in \mathbb{R}^{|S_{k-1}| \times |S|}$. Then, by Lemma 3.10, when $S = S_{k-1} \backslash \{j\}$, we have

$$
\left( \mathbf{X}_S^T \mathbf{X}_S \right)^{-1} = \mathbf{P}_j^T \left( \mathbf{X}_{S_{k-1}}^T \mathbf{X}_{S_{k-1}} \right)^{-1} \mathbf{P}_j - \frac{\mathbf{P}_j^T \left( \mathbf{X}_{S_{k-1}}^T \mathbf{X}_{S_{k-1}} \right)^{-1} \mathbf{e}_j \mathbf{e}_j^T \left( \mathbf{X}_{S_{k-1}}^T \mathbf{X}_{S_{k-1}} \right)^{-1} \mathbf{P}_j}{\mathbf{e}_j^T \left( \mathbf{X}_{S_{k-1}}^T \mathbf{X}_{S_{k-1}} \right)^{-1} \mathbf{e}_j},
$$

and $\mathbf{X}_S = \mathbf{X}_{S_{k-1}} \mathbf{P}_j$.

Therefore, problem (Q2) is equivalent to

$$
\underset{j \in S_{k-1}}{\mathrm{argmax}} \; \mathbf{y}^T \mathbf{X}_{S_{k-1}} \mathbf{P}_j \mathbf{P}_j^T \left( \mathbf{X}_{S_{k-1}}^T \mathbf{X}_{S_{k-1}} \right)^{-1} \mathbf{P}_j \mathbf{P}_j^T \mathbf{X}_{S_{k-1}}^T \mathbf{y}
$$

$$
- \frac{\mathbf{y}^T \mathbf{X}_{S_{k-1}} \mathbf{P}_j \mathbf{P}_j^T \left( \mathbf{X}_{S_{k-1}}^T \mathbf{X}_{S_{k-1}} \right)^{-1} \mathbf{e}_j \mathbf{e}_j^T \left( \mathbf{X}_{S_{k-1}}^T \mathbf{X}_{S_{k-1}} \right)^{-1} \mathbf{P}_j \mathbf{P}_j^T \mathbf{X}_{S_{k-1}}^T \mathbf{y}}{\mathbf{e}_j^T \left( \mathbf{X}_{S_{k-1}}^T \mathbf{X}_{S_{k-1}} \right)^{-1} \mathbf{e}_j}
$$

$$
= \underset{j \in S_{k-1}}{\mathrm{argmax}} \; \mathbf{y}^T \mathbf{X}_{S_{k-1}} \mathbf{P}_j \mathbf{P}_j^T \left( \left( \mathbf{X}_{S_{k-1}}^T \mathbf{X}_{S_{k-1}} \right)^{-1} - \frac{\left( \mathbf{X}_{S_{k-1}}^T \mathbf{X}_{S_{k-1}} \right)^{-1} \mathbf{e}_j \mathbf{e}_j^T \left( \mathbf{X}_{S_{k-1}}^T \mathbf{X}_{S_{k-1}} \right)^{-1}}{\mathbf{e}_j^T \left( \mathbf{X}_{S_{k-1}}^T \mathbf{X}_{S_{k-1}} \right)^{-1} \mathbf{e}_j} \right) \mathbf{P}_j \mathbf{P}_j^T \mathbf{X}_{S_{k-1}}^T \mathbf{y}
$$

$$
= \underset{j \in S_{k-1}}{\mathrm{argmax}} \; \mathbf{y}^T \mathbf{X}_{S_{k-1}} \left( \mathbf{I} - \mathbf{e}_j \mathbf{e}_j^T \right) \left( \mathbf{C}_{k-1} - \frac{\mathbf{C}_{k-1} \mathbf{e}_j \mathbf{e}_j^T \mathbf{C}_{k-1}}{\mathbf{e}_j^T \mathbf{C}_{k-1} \mathbf{e}_j} \right) \left( \mathbf{I} - \mathbf{e}_j \mathbf{e}_j^T \right) \mathbf{X}_{S_{k-1}}^T \mathbf{y}. \quad (\text{since } \mathbf{P}_j \mathbf{P}_j^T = \mathbf{I} - \mathbf{e}_j \mathbf{e}_j^T.)
$$

$$
\tag{17}
$$

$\qquad \square$

## D. Proof of Remark 3.13

*Proof.* When $\mathbf{X}_j$ and $\mathbf{X}_{S_{k-1}\setminus\{j\}}$ are orthogonal, we have $\mathbf{P}_j\left(\mathbf{X}_{S_{k-1}}^T\mathbf{X}_{S_{k-1}}\right)^{-1}\mathbf{e}_j = 0$. Therefore, the objective-based elimination criterion (10) can be reformulated as

$$(10) = \mathbf{y}^T\mathbf{X}_{S_{k-1}}\mathbf{P}_j\mathbf{P}_j^T\left(\mathbf{X}_{S_{k-1}}^T\mathbf{X}_{S_{k-1}}\right)^{-1}\mathbf{P}_j\mathbf{P}_j^T\mathbf{X}_{S_{k-1}}^T\mathbf{y}. \tag{18}$$

Noticing that $\mathbf{P}_j\mathbf{P}_j^T = \mathbf{I} - \mathbf{e}_j\mathbf{e}_j^T$, we can rewrite (18) as

$$
\begin{aligned}
(18) &= \mathbf{y}^T\mathbf{X}_{S_{k-1}}\left(\mathbf{I}-\mathbf{e}_j\mathbf{e}_j^T\right)\left(\mathbf{X}_{S_{k-1}}^T\mathbf{X}_{S_{k-1}}\right)^{-1}\left(\mathbf{I}-\mathbf{e}_j\mathbf{e}_j^T\right)\mathbf{X}_{S_{k-1}}^T\mathbf{y} \\
&= \left(\mathbf{y}^T\mathbf{X}_{S_{k-1}}\left(\mathbf{I}-\mathbf{e}_j\mathbf{e}_j^T\right)\left(\mathbf{X}_{S_{k-1}}^T\mathbf{X}_{S_{k-1}}\right)^{-1}\right)\left(\mathbf{X}_{S_{k-1}}^T\mathbf{X}_{S_{k-1}}\right)\left(\left(\mathbf{X}_{S_{k-1}}^T\mathbf{X}_{S_{k-1}}\right)^{-1}\left(\mathbf{I}-\mathbf{e}_j\mathbf{e}_j^T\right)\mathbf{X}_{S_{k-1}}^T\mathbf{y}\right). \tag{19}
\end{aligned}
$$

It's also observed that $\left(\mathbf{X}_{S_{k-1}}^T\mathbf{X}_{S_{k-1}}\right)^{-1}\left(\mathbf{I}-\mathbf{e}_j\mathbf{e}_j^T\right)\mathbf{X}_{S_{k-1}}^T\mathbf{y} \overset{(*)}{=} \left(\mathbf{I}-\mathbf{e}_j\mathbf{e}_j^T\right)\left(\mathbf{X}_{S_{k-1}}^T\mathbf{X}_{S_{k-1}}\right)^{-1}\mathbf{X}_{S_{k-1}}^T\mathbf{y} = \left(\mathbf{I}-\mathbf{e}_j\mathbf{e}_j^T\right)\boldsymbol{\beta}^{k-1}$, where $(*)$ holds because $\mathbf{X}_j$ is orthogonal to $\mathbf{X}_{S_{k-1}\setminus\{j\}}$. Therefore, (19) can be further expressed as follows:

$$(19) = \left(\boldsymbol{\beta}^{k-1}\right)^T\left(\mathbf{I}-\mathbf{e}_j\mathbf{e}_j^T\right)\left(\mathbf{X}_{S_{k-1}}^T\mathbf{X}_{S_{k-1}}\right)\left(\mathbf{I}-\mathbf{e}_j\mathbf{e}_j^T\right)\boldsymbol{\beta}^{k-1} = C - \left(\mathbf{X}_j^T\mathbf{X}_j\right)\left(\beta_j^{k-1}\right)^2. \tag{20}$$

Therefore, we have $(10) \iff \underset{j\in S_{k-1}}{\operatorname{argmax}}\ C - \left(\mathbf{X}_j^T\mathbf{X}_j\right)\left(\beta_j^{k-1}\right)^2 \iff \underset{j\in S_{k-1}}{\operatorname{argmin}}\ \left(\mathbf{X}_j^T\mathbf{X}_j\right)\left(\beta_j^{k-1}\right)^2$ when $\mathbf{X}_j \perp \mathbf{X}_{S_{k-1}\setminus\{j\}}, \forall j = 1,\cdots,|S_{k-1}|$. Thus, when $\mathbf{X}_{S_{k-1}}$ is a column-orthogonal matrix, the new criterion (10) reduces to minimizing the Wald-T statistics (4).

$\square$

## E. Enhanced Algorithms

### E.1. Optimal Pursuit

This subsection presents the algorithmic workflow of Optimal Pursuit (OP).

---

**Algorithm 1** OP: Optimal Pursuit

---

1: **Input:** Design matrix $\mathbf{X}$, response vector $\mathbf{y}$, sparsity level $K$, and residual tolerance $\epsilon$.
2: **Output:** Support set $\hat{S}$ and sparse solution $\hat{\boldsymbol{\beta}}$.
3: Initialize residual $\mathbf{r}^1 = \mathbf{y}$, support set $S_0 = \emptyset$, $\boldsymbol{\beta}^0 = \mathbf{0}$, $\mathbf{H}_0 = \mathbf{0}_{n\times n}$, and iteration counter $k = 1$.
4: **repeat**
5:     Compute the objective-based criterion (8) using $\mathbf{r}^k$, $\mathbf{H}_{k-1}$ and identify the optimal index $j$.     // Find the basis maximizing the objective reduction.
6:     Update support set: $S_k = S_{k-1} \cup \{j\}$.
7:     Update $\mathbf{C}_k = \left(\mathbf{X}_{S_k}^T\mathbf{X}_{S_k}\right)^{-1}$, and $\mathbf{H}_k = \mathbf{X}_{S_k}\mathbf{C}_k\mathbf{X}_{S_k}^T$.     // To be used in step $k+1$ as explained in Remark 3.7.
8:     Solve least-squares problem using $\boldsymbol{\beta}_{S_k}^k = \mathbf{C}_k\mathbf{X}_{S_k}^T\mathbf{y}$.     // Estimate coefficients on active set.
9:     Update $\boldsymbol{\beta}^k = \boldsymbol{\beta}_{S_k\cup S_k^c}^k$, where $\boldsymbol{\beta}_{S_k^c}^k = \mathbf{0}$.
10:     Update residual: $\mathbf{r}^{k+1} = \mathbf{y} - \mathbf{X}\boldsymbol{\beta}^k$.
11:     Update counter: $k = k + 1$.
12: **until** $\|\mathbf{r}^k\|_2 \le \epsilon$ or $k = K + 1$
13: $\hat{S} = S_{k-1}$, and $\hat{\boldsymbol{\beta}} = \boldsymbol{\beta}^{k-1}$.

---

### E.2. Compressive Sampling Optimal Pursuit

This subsection presents the algorithmic workflow of Compressive Sampling Optimal Pursuit (CoSaOP).

---

**Algorithm 2** CoSaOP: Compressive Sampling Optimal Pursuit

---

1: **Input:** Design matrix $\mathbf{X}$, response vector $\mathbf{y}$, sparsity level $K$, residual tolerance $\epsilon_1$, variation tolerance $\epsilon_2$, and maximum iteration count *Maxiter*.
2: **Output:** Support set $\hat{S}$ and sparse solution $\hat{\boldsymbol{\beta}}$.
3: Initialize residual $\mathbf{r}^1 = \mathbf{y}$, support set $S_0 = \emptyset$, $\boldsymbol{\beta}^0 = \mathbf{0}$, $\mathbf{H}_0 = \mathbf{0}_{n \times n}$, and iteration counter $k = 0$.
4: **repeat**
5:     Update counter: $k = k + 1$.
6:     Compute the objective-based criterion (8) using $\mathbf{r}^k$, $\mathbf{H}_{k-1}$ and identify the optimal $2K$ indexes $\Omega$.
7:     Update support set: $U_k = S_{k-1} \cup \Omega$.
8:     Update $\mathbf{C}_k^1 = \left(\mathbf{X}_{U_k}^T \mathbf{X}_{U_k}\right)^{-1}$.
9:     Compute the objective-based criterion (10) using $\mathbf{C}_k^1$ and identify the minimum $K$ indexes $S_k$.
10:     Update $\mathbf{C}_k = \left(\mathbf{X}_{S_k}^T \mathbf{X}_{S_k}\right)^{-1}$, and $\mathbf{H}_k = \mathbf{X}_{S_k} \mathbf{C}_k \mathbf{X}_{S_k}^T$.     // To be used in step $k+1$ as explained in Remark 3.7.
11:     Solve least-squares problem using $\boldsymbol{\beta}_{S_k}^k = \mathbf{C}_k \mathbf{X}_{S_k}^T \mathbf{y}$.
12:     Update $\boldsymbol{\beta}^k = \boldsymbol{\beta}_{S_k \cup S_k^c}^k$, where $\boldsymbol{\beta}_{S_k^c}^k = \mathbf{0}$.
13:     Update residual: $\mathbf{r}^{k+1} = \mathbf{y} - \mathbf{X}\boldsymbol{\beta}^k$.
14: **until** $\|\mathbf{r}^{k+1}\|_2 \leq \epsilon_1$ or $\|\boldsymbol{\beta}^k - \boldsymbol{\beta}^{k-1}\| \leq \epsilon_2$ or $k = $ *Maxiter*
15: $\hat{S} = S_k$, and $\hat{\boldsymbol{\beta}} = \boldsymbol{\beta}^k$ .

---

### E.3. Optimal Pursuit-Best-Subset Selection

This subsection presents the algorithmic workflow of Optimal Pursuit-Best-Subset Selection (OP-BESS) along with its sub-function, OP-Splicing.

---

**Algorithm 3** OP-BESS (Main Function): Optimal Pursuit-Best-Subset Selection

---

1: **Input:** Design matrix $\mathbf{X}$, response vector $\mathbf{y}$, a positive integer $k_{\max}$, and a threshold $\tau$.
2: **Output:** Support set $\hat{S}$, sparse solution $\hat{\boldsymbol{\beta}}$, and dual vector $\hat{\boldsymbol{d}}$.
3: Initialize $S^0 = \{j : \sum_{i=1}^p \mathbf{I}(|\frac{\mathbf{x}_j^T \mathbf{y}}{\sqrt{\mathbf{x}_j^T \mathbf{x}_j}}| \leq |\frac{\mathbf{x}_i^T \mathbf{y}}{\sqrt{\mathbf{x}_i^T \mathbf{x}_i}}| \leq K\}$, $\mathcal{I}^0 = (S^0)^c$, $\boldsymbol{\beta}_{\mathcal{I}^0}^0 = \mathbf{0}$, and $\boldsymbol{d}_{S^0}^0 = \mathbf{0}$.
4: Let $\mathbf{C}_0 = (\mathbf{X}_{S^0}^T \mathbf{X}_{S^0})^{-1}$.     // To be used in the next step as explained in Remark 3.7.
5: Compute initial $\boldsymbol{\beta}_{S^0}^0 = \mathbf{C}_0 \mathbf{X}_{S^0}^\top \mathbf{y}$, and $\boldsymbol{d}_{\mathcal{I}^0}^0 = \mathbf{X}_{\mathcal{I}^0}^\top (\mathbf{y} - \mathbf{X}\boldsymbol{\beta}^0)/n$.
6: **repeat**
7:     Update $(\boldsymbol{\beta}^{m+1}, \boldsymbol{d}^{m+1}, S^{m+1}, \mathcal{I}^{m+1}, \mathbf{C}_{m+1}) = \text{Splicing}(\boldsymbol{\beta}^m, \boldsymbol{d}^m, S^m, \mathcal{I}^m, \mathbf{C}_m, k_{\max}, \tau)$.
8: **until** $(S^{m+1}, \mathcal{I}^{m+1}) = (S^m, \mathcal{I}^m)$
9: **Return:** $(\hat{S}, \hat{\boldsymbol{\beta}}, \hat{\boldsymbol{d}}) = (S^{m+1}, \boldsymbol{\beta}^{m+1}, \boldsymbol{d}^{m+1})$.

---

**Algorithm 4** OP-Splicing (Sub-Function of OP-BESS): Update Model Parameters and Support Sets

---

1: **Input:** Current parameters obtained in step $m$: $\boldsymbol{\beta}$, $\boldsymbol{d}$, support sets $S, \mathcal{I}, \mathbf{C}_m$, maximum iterations $k_{\max}$, and threshold $\tau$.
2: **Output:** Updated parameters in iteration $m+1$: $\hat{\boldsymbol{\beta}}, \hat{\boldsymbol{d}}, \hat{S}, \hat{\mathcal{I}}$ and $\mathbf{C}_{m+1}$.
3: Initialize loss $L_0 = L = \|\mathbf{y} - \mathbf{X}\boldsymbol{\beta}\|_2^2/(2n)$.
4: Compute the objective-based criteria $\xi_j = (10)$ and $\zeta_j = (8)$ using $\mathbf{C}_m$ for $j = 1, \ldots, p$.
5: **for** $k = 1, 2, \ldots, k_{\max}$ **do**
6:     Update support sets:

$$S_k = \{j \in S : \sum_{i \in S} \mathbf{I}(\xi_j \leq \xi_i) \leq k\},$$

$$\mathcal{I}_k = \{j \in \mathcal{I} : \sum_{i \in \mathcal{I}} \mathbf{I}(\zeta_j \leq \zeta_i) \leq k\}.$$

7:     Let $\tilde{S}_k = (S \setminus S_k) \cup \mathcal{I}_k$, $\quad \tilde{\mathcal{I}}_k = (\mathcal{I} \setminus \mathcal{I}_k) \cup S_k$, and $\tilde{\mathbf{C}} = \left(\mathbf{X}_{\tilde{S}_k}^T \mathbf{X}_{\tilde{S}_k}\right)^{-1}$.     // To be used in the next step as explained in Remark 3.7.
8:     Solve least-squares problem for $\tilde{\boldsymbol{\beta}}_{\tilde{S}_k}$ and compute $\tilde{\boldsymbol{d}}$:

$$\tilde{\boldsymbol{\beta}}_{\tilde{S}_k} = \tilde{\mathbf{C}} \mathbf{X}_{\tilde{S}_k}^T \mathbf{y}, \ \ \tilde{\boldsymbol{\beta}}_{\tilde{\mathcal{I}}_k} = \mathbf{0}.$$

$$\tilde{\boldsymbol{d}} = \mathbf{X}^T \left(\mathbf{y} - \mathbf{X}\tilde{\boldsymbol{\beta}}/n\right).$$

9:     Update loss: $\mathcal{L}_n(\tilde{\boldsymbol{\beta}}) = \|\mathbf{y} - \mathbf{X}\tilde{\boldsymbol{\beta}}\|_2^2/(2n)$.

10:     **if** $L > \mathcal{L}_n(\tilde{\boldsymbol{\beta}})$ **then**
11:         Update $L = \mathcal{L}_n(\tilde{\boldsymbol{\beta}})$, $\hat{S} = \tilde{S}_k$, $\hat{\mathcal{I}} = \tilde{\mathcal{I}}_k$, $\hat{\boldsymbol{\beta}} = \tilde{\boldsymbol{\beta}}$, and $\mathbf{C}_{m+1} = \tilde{\mathbf{C}}$.
12:     **end if**
13: **end for**
14: **if** $L_0 - L < \tau$ **then**
15:     Return current parameters without updates.
16: **end if**

## F. Proof of Lemma 4.4

**Assumption.** *(Align with or is similar to those in (Tropp & Gilbert, 2007; Needell & Tropp, 2009; Zhu et al., 2020).)*
*1. $\boldsymbol{\beta}^*$ is $K$-sparse.*
*2. $\mathbf{X}$ satisfies Restricted Isometry Property (RIP) as in (Needell & Tropp, 2009). For definition of restricted isometry constant $\delta_r$, please refer to (Needell & Tropp, 2009).*
*3. $\mathbf{X}$ is column-normalized, i.e., $\mathbf{X}_j^T \mathbf{X}_j = 1$.*

*Proof.* The elements selected in $\Omega$ can be viewed as the $2K$ largest entries (in magnitude) of the vector $\mathbf{q} = \mathbf{D}\mathbf{X}^T \mathbf{r}^k$, where $\mathbf{D}$ is a diagonal matrix satisfying:

$$\mathbf{D}_{jj} = \begin{cases} \dfrac{1}{\sqrt{\mathbf{X}_j^T \left(\mathbf{I} - \mathbf{X}_{S_{k-1}} \left(\mathbf{X}_{S_{k-1}}^T \mathbf{X}_{S_{k-1}}\right)^{-1} \mathbf{X}_{S_{k-1}}^T\right)\mathbf{X}_j}}, & j \in S_{k-1}^c, \\[3mm] 1, & j \in S_{k-1}. \end{cases}$$

Notation: $\mathbf{X}_j^T \left(\mathbf{I} - \mathbf{X}_{S_{k-1}} \left(\mathbf{X}_{S_{k-1}}^T \mathbf{X}_{S_{k-1}}\right)^{-1} \mathbf{X}_{S_{k-1}}^T\right) \mathbf{X}_j = \mathbf{X}_j^T \left(\mathbf{I} - \mathbf{X}_{S_{k-1}} \left(\mathbf{X}_{S_{k-1}}^T \mathbf{X}_{S_{k-1}}\right)^{-1} \mathbf{X}_{S_{k-1}}^T\right)^2 \mathbf{X}_j$. When $j \in S_{k-1}^c$, $\mathbf{X}_j^T \left(\mathbf{I} - \mathbf{X}_{S_{k-1}} \left(\mathbf{X}_{S_{k-1}}^T \mathbf{X}_{S_{k-1}}\right)^{-1} \mathbf{X}_{S_{k-1}}^T\right)^2 \mathbf{X}_j > 0$. Therefore, $\mathbf{D}$ is well-defined.

Let $R = \text{supp}\left(\boldsymbol{\beta}^{k-1} - \boldsymbol{\beta}^*\right)$. According to CoSaOP algorithm, the set $R$ contains at most $2K$ nonzero elements. By the definition of $\Omega$, it holds that $\|\mathbf{q}|_R\|_2 \leq \|\mathbf{q}|_\Omega\|_2$. Then, we have

$$\left\|\mathbf{q}|_{R \setminus \Omega}\right\|_2 \leq \left\|\mathbf{q}|_{\Omega \setminus R}\right\|_2.$$

By (Needell & Tropp, 2009) (Proposition 3.1 and 3.2), for $j \in S_{k-1}^c$, it holds that

$$\mathbf{X}_j^T \mathbf{X}_{S_{k-1}} \left(\mathbf{X}_{S_{k-1}}^T \mathbf{X}_{S_{k-1}}\right)^{-1} \mathbf{X}_{S_{k-1}}^T \mathbf{X}_j \leq \left\|\mathbf{X}_{S_{k-1}}^T \mathbf{X}_j\right\|_2 \left\|\left(\mathbf{X}_{S_{k-1}}^T \mathbf{X}_{S_{k-1}}\right)^{-1} \mathbf{X}_{S_{k-1}}^T \mathbf{X}_j\right\|_2$$

$$\leq \delta_{K+1} \cdot \frac{1}{\sqrt{1 - \delta_K}}.$$

Therefore,

$$1 - \frac{\delta_{K+1}}{\sqrt{1 - \delta_K}} \leq \mathbf{X}_j^T \left(\mathbf{I} - \mathbf{X}_{S_{k-1}} \left(\mathbf{X}_{S_{k-1}}^T \mathbf{X}_{S_{k-1}}\right)^{-1} \mathbf{X}_{S_{k-1}}^T\right) \mathbf{X}_j \leq 1,$$

i.e.,

$$1 \leq \mathbf{D}_{jj} \leq \frac{1}{\sqrt{1 - \frac{\delta_{K+1}}{\sqrt{1 - \delta_K}}}},$$

which implies

$$\|\mathbf{D}\|_2 \leq \frac{1}{\sqrt{1 - \frac{\delta_{K+1}}{\sqrt{1 - \delta_K}}}}. \tag{21}$$

Let $\mathbf{D}|_{\Omega\setminus R}$ denote the submatrix of $\mathbf{D}$ restricted to the index set $\Omega\setminus R$, with dimensions $|\Omega\setminus R| \times |\Omega\setminus R|$. Therefore, by (Needell & Tropp, 2009) (Proposition 3.1 and Corollary 3.3) and (21), it holds that

$$
\begin{aligned}
\left\|\mathbf{q}|_{\Omega\setminus R}\right\|_2 &= \left\|\mathbf{D}|_{\Omega\setminus R} \cdot \mathbf{X}_{\Omega\setminus R}^T \mathbf{r}^k\right\|_2 \\
&\leq \left\|\mathbf{D}|_{\Omega\setminus R}\right\|_2 \left\|\mathbf{X}_{\Omega\setminus R}^T \mathbf{r}^k\right\|_2 \\
&\leq \frac{1}{\sqrt{1 - \frac{\delta_{K+1}}{\sqrt{1-\delta_K}}}} \left\|\mathbf{X}_{\Omega\setminus R}^T \left(\mathbf{X}\left(\boldsymbol{\beta}^* - \boldsymbol{\beta}^{k-1}\right) + \boldsymbol{\epsilon}\right)\right\|_2 \\
&\leq \frac{1}{\sqrt{1 - \frac{\delta_{K+1}}{\sqrt{1-\delta_K}}}} \left(\left\|\mathbf{X}_{\Omega\setminus R}^T \mathbf{X}\left(\boldsymbol{\beta}^* - \boldsymbol{\beta}^{k-1}\right)\right\|_2 + \left\|\mathbf{X}_{\Omega\setminus R}^T \boldsymbol{\epsilon}\right\|_2\right) \\
&\leq \frac{1}{\sqrt{1 - \frac{\delta_{K+1}}{\sqrt{1-\delta_K}}}} \left(\delta_{4K} \left\|\boldsymbol{\beta}^* - \boldsymbol{\beta}^{k-1}\right\|_2 + \sqrt{1 + \delta_{2K}}\|\boldsymbol{\epsilon}\|_2\right).
\end{aligned}
$$

By the definition of the matrix $\mathbf{D}$ and $\mathbf{D}_{jj} \geq 1$, for $j \in S_{k-1}^c$, it follows that for any $\mathbf{u} \in \mathbb{R}^n$,

$$
\|\mathbf{D}\mathbf{u}\|_2^2 = \sum_{j=1}^n \mathbf{D}_{jj}^2 \mathbf{u}_j^2 \geq \sum_{j=1}^n \mathbf{u}_j^2 = \|\mathbf{u}\|_2^2. \tag{22}
$$

Similarly, according to (Needell & Tropp, 2009) (Proposition 3.1 and Corollary 3.3) and (22),

$$
\begin{aligned}
\left\|\mathbf{q}|_{R\setminus\Omega}\right\|_2 &= \left\|\mathbf{D}|_{R\setminus\Omega} \cdot \mathbf{X}_{R\setminus\Omega}^T \mathbf{r}^k\right\|_2 \\
&\geq \left\|\mathbf{X}_{R\setminus\Omega}^T \mathbf{r}^k\right\|_2 \\
&= \left\|\mathbf{X}_{R\setminus\Omega}^T \left(\mathbf{X}\left(\boldsymbol{\beta}^* - \boldsymbol{\beta}^{k-1}\right) + \boldsymbol{\epsilon}\right)\right\|_2 \\
&\geq \left\|\mathbf{X}_{R\setminus\Omega}^T \mathbf{X}\left(\boldsymbol{\beta}^* - \boldsymbol{\beta}^{k-1}\right)|_{R\setminus\Omega}\right\|_2 - \left\|\mathbf{X}_{R\setminus\Omega}^T \mathbf{X}\left(\boldsymbol{\beta}^* - \boldsymbol{\beta}^{k-1}\right)|_{\Omega}\right\|_2 - \|\mathbf{X}_{R\setminus\Omega}^T \boldsymbol{\epsilon}\| \\
&\geq (1 - \delta_{2K}) \left\|\left(\boldsymbol{\beta}^* - \boldsymbol{\beta}^{k-1}\right)|_{R\setminus\Omega}\right\|_2 - \delta_{2K} \left\|\boldsymbol{\beta}^* - \boldsymbol{\beta}^{k-1}\right\|_2 - \sqrt{1 + \delta_{2K}}\|\boldsymbol{\epsilon}\|_2.
\end{aligned}
$$

Based on the definition of $R$, it holds that $\left(\boldsymbol{\beta}^* - \boldsymbol{\beta}^{k-1}\right)|_{R\setminus\Omega} = \left(\boldsymbol{\beta}^* - \boldsymbol{\beta}^{k-1}\right)|_{\Omega^c}$. Therefore, we have

$$
\begin{aligned}
&(1 - \delta_{2K}) \left\|\left(\boldsymbol{\beta}^* - \boldsymbol{\beta}^{k-1}\right)|_{\Omega^c}\right\|_2 - \delta_{2K} \left\|\boldsymbol{\beta}^* - \boldsymbol{\beta}^{k-1}\right\|_2 - \sqrt{1 + \delta_{2K}}\|\boldsymbol{\epsilon}\|_2 \\
&\leq \frac{1}{\sqrt{1 - \frac{\delta_{K+1}}{\sqrt{1-\delta_K}}}} \left(\delta_{4K} \left\|\boldsymbol{\beta}^* - \boldsymbol{\beta}^{k-1}\right\|_2 + \sqrt{1 + \delta_{2K}}\|\boldsymbol{\epsilon}\|_2\right).
\end{aligned}
$$

Then, it follows:

$$
\left\|\left(\boldsymbol{\beta}^* - \boldsymbol{\beta}^{k-1}\right)|_{\Omega^c}\right\|_2 \leq \frac{\left(\frac{\delta_{4K}}{\sqrt{1 - \frac{\delta_{K+1}}{\sqrt{1-\delta_K}}}} + \delta_{2K}\right) \left\|\boldsymbol{\beta}^* - \boldsymbol{\beta}^{k-1}\right\|_2 + \left(\frac{\sqrt{1+\delta_{2K}}}{\sqrt{1 - \frac{\delta_{K+1}}{\sqrt{1-\delta_K}}}} + \sqrt{1 + \delta_{2K}}\right)\|\boldsymbol{\epsilon}\|_2}{1 - \delta_{2K}}.
$$

By assumption $\delta_K \leq \delta_{K+1} \leq \delta_{2K} \leq \delta_{4K} \leq 0.1$, Lemma 4.4 is proved.

$\square$

## G. Proof of Lemma 4.5

*Proof.* The proof follows from Lemma 4.3 in (Needell & Tropp, 2009). Since $\text{supp}(\boldsymbol{\beta}^{k-1}) \subset U$, we find that

$$
\left\|\boldsymbol{\beta}^*|_{U^c}\right\|_2 = \left\|\left(\boldsymbol{\beta}^* - \boldsymbol{\beta}^{k-1}\right)|_{U^c}\right\|_2 \leq \left\|\left(\boldsymbol{\beta}^* - \boldsymbol{\beta}^{k-1}\right)|_{\Omega^c}\right\|_2.
$$

$\square$

# H. Proof of Lemma 4.6

*Proof.* The proof follows from Lemma 4.4 in (Needell & Tropp, 2009). □

# I. Proof of Lemma 4.7

**Assumption.** *(Following the three assumptions in Appendix F.)*
*Let $\tilde{\mathbf{a}}_{S_k}$ be defined as follows:*

$$\tilde{\mathbf{a}}_{S_k,j} = \begin{cases} \mathbf{a}_j, & \text{if } j \in S_k, \\ 0, & \text{if } j \in S_k^c, \end{cases}$$

*and let $\tilde{\mathbf{a}}_K$ denote the best $K$-sparse approximation of $\mathbf{a}$.*

*4. $supp(\boldsymbol{\beta}^*) \subseteq (S_k \triangle supp(\tilde{\mathbf{a}}_K))^c$, where $A \triangle B = (A \backslash B) \cup (B \backslash A)$.*

*Proof.*

$$\|\boldsymbol{\beta}^* - \tilde{\mathbf{a}}_{S_k}\|_2 \le \|\boldsymbol{\beta}^* - \tilde{\mathbf{a}}_K\|_2 + \|\tilde{\mathbf{a}}_K - \tilde{\mathbf{a}}_{S_k}\|_2,$$
$$\|\boldsymbol{\beta}^* - \tilde{\mathbf{a}}_K\|_2 \le \|\boldsymbol{\beta}^* - \mathbf{a}\|_2 + \|\mathbf{a} - \tilde{\mathbf{a}}_K\|_2 \le 2\|\boldsymbol{\beta}^* - \mathbf{a}\|_2,$$
$$\|\tilde{\mathbf{a}}_K - \tilde{\mathbf{a}}_{S_k}\|_2 = \left\|\tilde{\mathbf{a}}_{\text{supp}(\tilde{\mathbf{a}}_K) \triangle S_k}\right\|_2 \le \delta \|\boldsymbol{\beta}^* - \mathbf{a}\|_2 \quad \text{(by Assumption 4)}.$$

Since $\|\boldsymbol{\beta}^* - \mathbf{a}\|_2 = \left\|\tilde{\mathbf{a}}_{\text{supp}(\tilde{\mathbf{a}}_K) \triangle S_k}\right\|_2 + \left\|(\boldsymbol{\beta}^* - \mathbf{a})|_{(\text{supp}(\tilde{\mathbf{a}}_K) \triangle S_k)^c}\right\|_2 \ge \left\|\tilde{\mathbf{a}}_{\text{supp}(\tilde{\mathbf{a}}_K) \triangle S_k}\right\|_2$, when $\mathbf{X}_U$ is orthogonal, $\text{supp}(\tilde{\mathbf{a}}_K) \triangle S_k = \emptyset, \delta = 0$. When the orthogonality of $\mathbf{X}_U$ is weak, $\text{supp}(\tilde{\mathbf{a}}_K) \cap S_k = \emptyset, \delta = 1$. Therefore, $\|\boldsymbol{\beta}^* - \tilde{\mathbf{a}}_{S_k}\|_2 \le (2 + \delta)\|\boldsymbol{\beta}^* - \mathbf{a}\|_2$. □

# J. Proof of Lemma 4.8

*Proof.* The properties of least squares imply that $\|\mathbf{y} - \mathbf{X}\boldsymbol{\beta}^k\|_2 \le \|\mathbf{y} - \mathbf{X}\tilde{\mathbf{a}}_{S_k}\|_2$. Therefore, it holds that

$$\left\|\mathbf{X}\left(\boldsymbol{\beta}^* - \boldsymbol{\beta}^k\right) + \boldsymbol{\epsilon}\right\|_2 \le \|\mathbf{X}(\boldsymbol{\beta}^* - \tilde{\mathbf{a}}_{S_k}) + \boldsymbol{\epsilon}\|_2,$$

which leads to

$$\left\|\mathbf{X}\left(\boldsymbol{\beta}^* - \boldsymbol{\beta}^k\right)\right\|_2 - \|\boldsymbol{\epsilon}\|_2 \le \|\mathbf{X}(\boldsymbol{\beta}^* - \tilde{\mathbf{a}}_{S_k})\|_2 + \|\boldsymbol{\epsilon}\|_2,$$

i.e.,

$$\left\|\mathbf{X}\left(\boldsymbol{\beta}^* - \boldsymbol{\beta}^k\right)\right\|_2 \le \|\mathbf{X}(\boldsymbol{\beta}^* - \tilde{\mathbf{a}}_{S_k})\|_2 + 2\|\boldsymbol{\epsilon}\|_2.$$

Suppose $E = \text{supp}(\boldsymbol{\beta}^*) \cup S_k, |E| \le 2K$, then

$$\left\|\mathbf{X}\left(\boldsymbol{\beta}^* - \boldsymbol{\beta}^k\right)\right\|_2^2 = \left\|\mathbf{X}_E\left(\boldsymbol{\beta}^* - \boldsymbol{\beta}^k\right)|_E\right\|_2^2$$
$$= \left(\boldsymbol{\beta}^* - \boldsymbol{\beta}^k\right)^T|_E \mathbf{X}_E^T\mathbf{X}_E\left(\boldsymbol{\beta}^* - \boldsymbol{\beta}^k\right)|_E$$
$$\ge \lambda_{\min}\left(\mathbf{X}_E^T\mathbf{X}_E\right)\left\|\boldsymbol{\beta}^* - \boldsymbol{\beta}^k\right\|_2^2.$$

Similarly, we have

$$\|\mathbf{X}(\boldsymbol{\beta}^* - \tilde{\mathbf{a}}_{S_k})\|_2^2 \le \lambda_{\max}\left(\mathbf{X}_E^T\mathbf{X}_E\right)\|\boldsymbol{\beta}^* - \tilde{\mathbf{a}}_{S_k}\|_2^2.$$

Therefore,

$$\left\| \boldsymbol{\beta}^* - \boldsymbol{\beta}^k \right\|_2 \le \sqrt{\frac{\lambda_{\max}\left(\mathbf{X}_E^T \mathbf{X}_E\right)}{\lambda_{\min}\left(\mathbf{X}_E^T \mathbf{X}_E\right)}} \left\| \boldsymbol{\beta}^* - \tilde{\mathbf{a}}_{S_k} \right\|_2 + \frac{2}{\sqrt{\lambda_{\min}\left(\mathbf{X}_E^T \mathbf{X}_E\right)}} \|\boldsymbol{\epsilon}\|_2$$

$$\le \sqrt{\frac{1 + \delta_{2K}}{1 - \delta_{2K}}} \left\| \boldsymbol{\beta}^* - \tilde{\mathbf{a}}_{S_k} \right\|_2 + \frac{2}{\sqrt{1 - \delta_{2K}}} \|\boldsymbol{\epsilon}\|_2 \quad \text{(by Proposition 3.1 in (Needell \& Tropp, 2009))}$$

$$\le 1.106 \left\| \boldsymbol{\beta}^* - \tilde{\mathbf{a}}_{S_k} \right\|_2 + 2.109 \|\boldsymbol{\epsilon}\|_2.$$

$\square$

## K. Proof of Theorem 4.9

*Proof.* Based on Lemma 4.4-4.8, we can derive the linear convergence rate of CoSaOP as follows:

$$\left\| \boldsymbol{\beta}^* - \boldsymbol{\beta}^k \right\|_2 \le 1.106 \left\| \boldsymbol{\beta}^* - \tilde{\mathbf{a}}_{S_k} \right\|_2 + 2.109 \|\boldsymbol{\epsilon}\|_2 \quad \text{(Lemma 4.8)}$$

$$\le 1.106(2 + \delta) \left\| \boldsymbol{\beta}^* - \mathbf{a} \right\|_2 + 2.109 \|\boldsymbol{\epsilon}\|_2 \quad \text{(Lemma 4.7)}$$

$$\le 1.106(2 + \delta)\left(1.112 \left\| \boldsymbol{\beta}^*|_{U^c} \right\|_2 + 1.06\|\boldsymbol{\epsilon}\|_2\right) + 2.109 \|\boldsymbol{\epsilon}\|_2 \quad \text{(Lemma 4.6)}$$

$$\le 1.106(2 + \delta)\left(1.112 \left\| \left(\boldsymbol{\beta}^* - \boldsymbol{\beta}^{k-1}\right)\big|_{\Omega^c} \right\|_2 + 1.06\|\boldsymbol{\epsilon}\|_2\right) + 2.109 \|\boldsymbol{\epsilon}\|_2 \quad \text{(Lemma 4.5)}$$

$$= 1.23(2 + \delta) \left\| \left(\boldsymbol{\beta}^* - \boldsymbol{\beta}^{k-1}\right)\big|_{\Omega^c} \right\|_2 + (2.109 + 1.173(2 + \delta)\|\boldsymbol{\epsilon}\|_2)$$

$$\le 1.23(2 + \delta)\left(0.2353 \left\| \boldsymbol{\beta}^* - \boldsymbol{\beta}^{k-1} \right\|_2 + 2.4\|\boldsymbol{\epsilon}\|_2\right) + (2.109 + 1.173(2 + \delta)\|\boldsymbol{\epsilon}\|_2) \quad \text{(Lemma 4.4)}$$

$$\le 0.869 \left\| \boldsymbol{\beta}^* - \boldsymbol{\beta}^{k-1} \right\|_2 + 14.482\|\boldsymbol{\epsilon}\|_2.$$

$\square$

## L. Proof of Theorem 4.10

*Proof.* When $i \in S$, $\rho = \frac{|\mathbf{X}_i^T \mathbf{X}_j|}{\|\mathbf{X}_i\|_2 \|\mathbf{X}_j\|_2}$, we have

$$\left\| \left(\mathbf{I} - \mathbf{X}_S\left(\mathbf{X}_S^T \mathbf{X}_S\right)^{-1}\mathbf{X}_S^T\right)\mathbf{X}_j \right\|_2 \le \sqrt{1 - \rho}\|\mathbf{X}_j\|_2. \tag{23}$$

Hence, for classical correlation-based selection criterion (3),

$$\frac{|\mathbf{r}^{k^T}\mathbf{X}_j|}{\|\mathbf{X}_j\|_2} = \frac{\mathbf{r}^{k^T}\left(\mathbf{I} - \mathbf{X}_S\left(\mathbf{X}_S^T \mathbf{X}_S\right)^{-1}\mathbf{X}_S^T\right)\mathbf{X}_j}{\|\mathbf{X}_j\|_2}$$

$$\le \frac{\|\mathbf{r}^k\|_2\|\mathbf{X}_j\|_2\sqrt{1 - \rho^2}}{\|\mathbf{X}_j\|_2}$$

$$= \sqrt{1 - \rho^2}\|\mathbf{r}^k\|_2.$$

While for proposed criterion,

$$\frac{\left(\mathbf{r}^{k^T}\mathbf{X}_j\right)^2}{\mathbf{X}_j^T\left(\mathbf{I} - \mathbf{X}_S\left(\mathbf{X}_S^T \mathbf{X}_S\right)^{-1}\mathbf{X}_S^T\right)\mathbf{X}_j} = \frac{\left(\mathbf{r}^{k^T}\mathbf{X}_j\right)^2}{\mathbf{X}_j^T\left(\mathbf{I} - \mathbf{X}_S\left(\mathbf{X}_S^T \mathbf{X}_S\right)^{-1}\mathbf{X}_S^T\right)^2\mathbf{X}_j} \ge \frac{1}{1 - \rho^2}\left(\frac{\mathbf{r}^{k^T}\mathbf{X}_j}{\|\mathbf{X}_j\|_2}\right)^2. \tag{24}$$

$\square$

# M. Proof of Theorem 4.11

*Proof.* The proof of Theorem 4.11 (1) and (2) will be given below.

(1) Since

$$
\begin{aligned}
\textit{objective-based criterion (10)} &= \mathbf{y}^T \mathbf{X}_S \left( \mathbf{X}_S^T \mathbf{X}_S \right)^{-1} \mathbf{X}_S^T \mathbf{y} \\
&= -||\mathbf{y} - \mathbf{X}_S \boldsymbol{\beta}\big|_S||_2^2 + ||\mathbf{y}||_2^2 \\
&\leq ||\mathbf{y}||_2^2,
\end{aligned}
\tag{25}
$$

the upper bound of the objective-based criterion (10) is $||\mathbf{y}||_2^2$.

Now consider that

$$
\mathbf{y} = \mathbf{X}\boldsymbol{\beta}^* + \boldsymbol{\epsilon} = \mathbf{X}_{S^*} \boldsymbol{\beta}^* \big|_{S^*} + \boldsymbol{\epsilon}.
\tag{26}
$$

If $S^* \subset S$, then for all $j_m \in S \backslash S^*$, $S^* \subset S \backslash \{j_m\}$. By (25),

$$
\begin{aligned}
\textit{objective-based criterion (10) for } j_m &= \mathbf{y}^T \mathbf{X}_{S \backslash \{j_m\}} \left( \mathbf{X}_{S \backslash \{j_m\}}^T \mathbf{X}_{S \backslash \{j_m\}} \right)^{-1} \mathbf{X}_{S \backslash \{j_m\}}^T \mathbf{y} \\
&= -||\mathbf{y} - \mathbf{X}_{S \backslash \{j_m\}} \boldsymbol{\beta}\big|_{S \backslash \{j_m\}}||_2^2 + ||\mathbf{y}||_2^2 \\
&\geq -||\mathbf{y} - \mathbf{X}_{S^*} \boldsymbol{\beta}^* \big|_{S^*}||_2^2 + ||\mathbf{y}||_2^2 \\
&= ||\mathbf{y}||_2^2 - ||\boldsymbol{\epsilon}||_2^2.
\end{aligned}
$$

Hence, for all $j_m \in S \backslash S^*$, the lower bound of the objective-based criterion (10) for $j_m$ is $||\mathbf{y}||_2^2 - ||\boldsymbol{\epsilon}||_2^2$.

In noiseless scenario, for all $j_m \in S \backslash S^*$, the objective-based criterion (10) for $j_m$ achieves the upper bound. Hence,

$$
j_m \in \operatorname{argmax}_{j \in S} \textit{ objective-based criterion (10)}.
$$

(2) Consider a simple counterexample where the T-statistic fails in the presence of highly correlated features.

Let

$$
\mathbf{X} = \begin{pmatrix} 0.2 & 0 & 0 \\ 0 & 0.8 & 0.9 \\ 0 & 0.1 & 0.1 \end{pmatrix}.
$$

and $\mathbf{y} = (0.2, 0.85, 0.1)^T$. Consider the best subset selection with $K = 2$. The ground truth is as follows:

*Table 2.* A simple counterexample.

| Subset | Feature 1 & 2 | Feature 2 & 3 | Feature 1 & 3 |
|---|---|---|---|
| Residual Norm | 0.0062 | 0.2 | 0.0055 |
| Best Subset | No | No | Yes |

Hence, when $K = 2$, feature $\mathbf{X}_1$ and $\mathbf{X}_3$ are the true features, and feature $\mathbf{X}_2$ is the spurious feature.

However, given $S = \{1, 2, 3\}$, the classic criterion (4) ($\sqrt{\mathbf{X}_i^T \mathbf{X}_i} |\boldsymbol{\beta}_i|$) for the features are: $T_1 = 0.2, T_2 = 0.403, T_3 = 0.453$. By minimizing classic criterion (4), feature $\mathbf{X}_1$ is eliminated.

The core issue lies in the fact that the spurious feature $\mathbf{X}_2$ is highly correlated with the important feature $\mathbf{X}_3$ in the true subset. Since classic criterion (4) considers only the individual significance of features while ignoring their interactions, the true feature $\mathbf{X}_1$ is erroneously discarded by the classic criterion (4).

However, for objective-based criterion (10), it will always identify $\mathbf{X}_p$ by Theorem 4.2 in the paper. The proof is complete. $\square$

# N. Explanations on Theorem 4.10 and Theorem 4.11

**Explanations on Theorem 4.10:**

Theorem 4.10 demonstrates that if features $i$ and $j$ in the true subset are correlated (violating the RIP condition), inequality (11) implies that as their correlation $\rho$ approaches 1, feature $j$ becomes increasingly unidentifiable under traditional correlation-based criterion (3) once feature $i$ has been selected. In contrast, (12) shows that when $\rho$ is large, since the first term's coefficient in RHS $1/(1-\rho^2)$ is large, the proposed criterion (8) exists a stable lower bound. **The larger the correlation $\rho$, the stronger discrepancy between traditional correlation-based criterion (3) and proposed criterion (8).** This significantly mitigates the impact of feature correlations on identifying $j$, making true features more reliably detectable.

**Explanations on Theorem 4.11:**

(1) The upper bound of the objective-based criterion (10) is $\|\mathbf{y}\|_2^2$. Theorem 4.11 (1) shows that when the current subset $S$ contains the true subset $S^*$, all features outside the true subset have a stable lower bound for their criterion (10), which is related to noise level. When the noise is relatively weak compared to the signal, this lower bound becomes larger, indicating that features outside the true subset are more easily identified by maximizing criterion (10) (ideally, larger values of criterion (10) for irrelevant features are preferred). In the noiseless case, all features outside the true subset can be identified by maximizing criterion (10). This result does not rely on any assumptions about feature correlations, and therefore remains valid even in scenarios with highly correlated features.

In Figure 3, all enhanced algorithms that incorporate the feature elimination criterion (10) show a clear improvement in performance as the SNR increases. This is also consistent with the theoretical results.

(2) In the best subset selection problem, a pseudo-correlation phenomenon may arise. Pseudo-correlated features refer to those features $\mathbf{X}_p$ that are highly correlated with critical features $\mathbf{X}_i$ in the true subset $S^*$, yet $\mathbf{X}_p$ itself does not belong to $S^*$. Once such features are selected, they are difficult to detect and remove by classical T-statistics based feature elimination criterion (4) but may be reliably identified by the proposed criterion (10). See Appendix M for example.

Theorem 4.10 and 4.11 demonstrate that our proposed criterion (8) effectively identifies truly important features even when they are strongly correlated, and criterion (10) reliably removes pseudo-correlated features that mimic true features. These theoretical insights are further validated by the experimental results in Appendix P, which confirm the superior performance of our criteria in high-correlation scenarios.

# O. Experiment on Computational Time

In this section, we report the computational time of each algorithm over 500 independent runs. While the new criteria introduce additional multiplication operations, resulting in slightly higher computational time compare to the original criteria [2], it remains within the same order of magnitude, as shown in Figure 5.

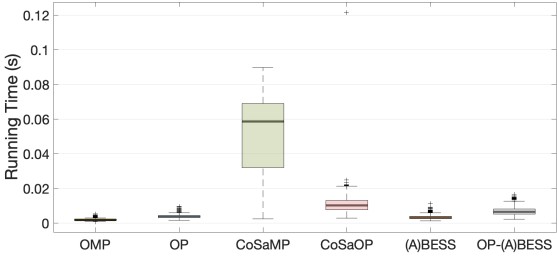

*Figure 5.* Running time for each algorithm over 500 independent runs.

---

[2]In Figure 5, CoSaMP requires more time than CoSaOP, since in this sparse recovery scenario, although CoSaOP performs additional matrix multiplications in a single iteration, it generally converges to the stopping criterion within just a few iterations. In contrast, CoSaMP requires significantly more iterations to converge, often reaching the maximum number of iterations before stopping, which results in a longer runtime.

## P. Experiment with High-dimensional Correlated Features and Small Noisy Samples

In this experiment, we conduct additional comparisons of the algorithm under extreme scenarios:

(a) **Small-sample rate and high-dimensional vectors:** $p = 2000$, with $n/p$ varying from 0.05 to 0.1.

(b) **High noise:** SNR[3] varies from 5 to 15.

(c) **Highly correlated features (RIP violated):** The covariance matrix of the row vectors of $\mathbf{X}$ follows a Toeplitz structure, where the correlation between position $i, j$ is $corr_{ij} = \rho^{|i-j|}$, with $\rho = 0.7$.

We generated sparse vectors with a sparsity level $K = 10$ for testing. Specifically, the sparse signal is block-sparse, comprising two blocks of five adjacent non-zero entries each. Combined with the Toeplitz covariance structure (where features closer in position exhibit higher correlations), this configuration ensures:

(1) **High correlation features within the true subset** (as stated in Theorem 4.10).

(2) **Many pseudo-features outside the true subset highly correlated with those in the true subset** (as in Theorem 4.11).

Phase transition diagrams illustrate the combined impact of varying sampling rates and SNR on algorithm performance (using NMSE as evaluation metric), where larger blue areas indicate stronger algorithm performance. The experimental results show that all algorithms enhanced with the new criteria exhibit significant improvements in phase transition capabilities compared to their original versions. This not only demonstrates the robust advantages of the new criteria in high-dimensional, low-SNR extreme scenarios but also validates the theories established in Theorem 4.10 and 4.11 through their marked improvements under high feature correlations.

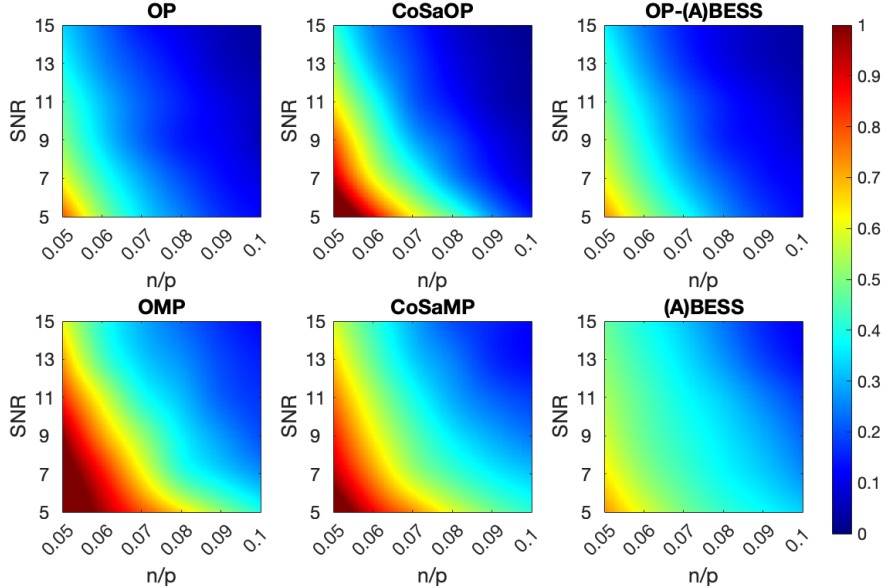

*Figure 6.* Phase transition with correlated features.

## Q. Reasons Why CoSaMP Fails in Sparse Regression

The main algorithmic flow of CoSaMP is as follows: it iteratively (1) selects $2K$ features, (2) solves a least squares problem on a large subset, and (3) prunes to $K$ coefficients. However, high feature correlation can cause significant errors in the final estimate from steps (2) and (3). We visualize the impact of feature correlation on CoSaMP's iterative process here.

---

[3]$SNR = 20 \log_{10} ||\mathbf{X}\boldsymbol{\beta}|| / ||\boldsymbol{\epsilon}||$.

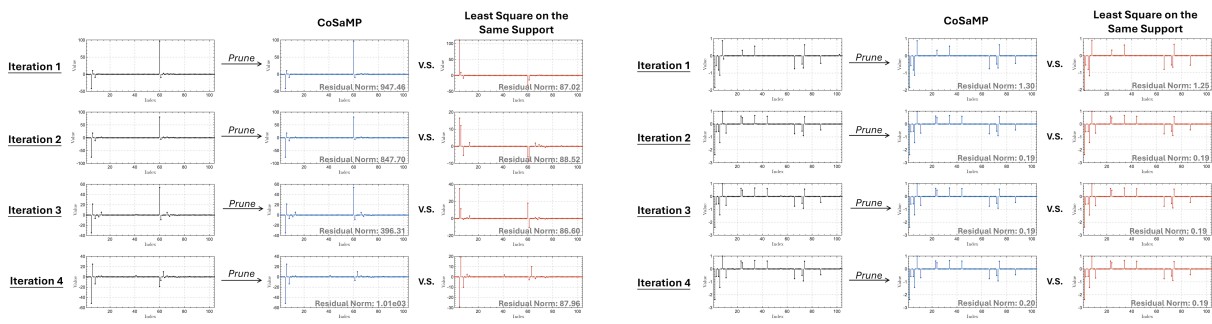

(a) CoSaMP fails when features are highly correlated    (b) CoSaMP succeeds when features are weakly correlated

*Figure 7.* Reasons why CoSaMP fails in sparse regression.

As shown in Figure 7a, on a regression dataset (Boston Housing) with highly correlated features, the pruned support set's direct coefficients (column 2) differ significantly from those after least squares estimation (column 3), with substantial residuals. CoSaMP, lacking least squares refinement, fails as the residuals grow with each iteration due to high feature correlation. This is evident in the residual curve evolution in Figure 8a.

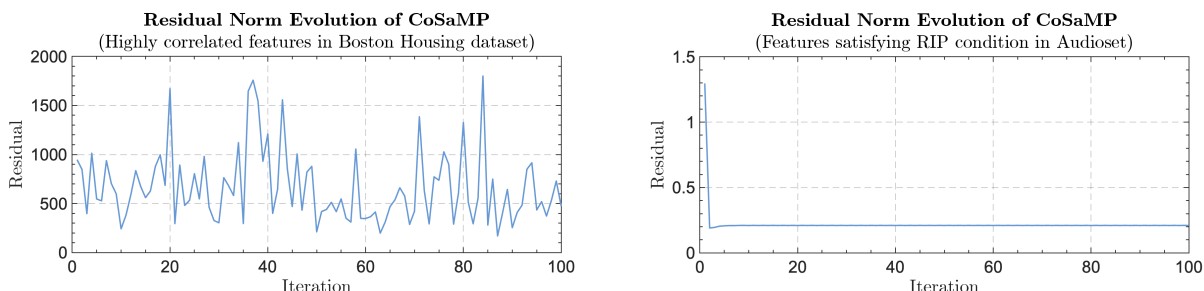

(a) Residual evolution of CoSaMP for highly-correlated features    (b) Residual evolution of CoSaMP for weakly-correlated features

*Figure 8.* Residual evolution of CoSaMP with different feature correlation cases.

In contrast, when features are weakly correlated (as shown in Figure 7b), the coefficients and residuals after pruning the large support set (column 2) and performing least squares (column 3) are nearly identical, leading to algorithm convergence, as shown in Figure 8b.

Therefore, CoSaMP is less suitable for scenarios where features are highly correlated or when $p$ is close to $3K$. The theoretical properties of this algorithm, as established in (Needell & Tropp, 2009), are also based on the assumption of weak feature correlation.

However, CoSaMP performs well in compressed sensing scenarios, particularly when using NMSE as the evaluation metric for audio data, where it can almost achieve the best results among classical algorithms.

## R. Cross-Validation Performance in Prediction

We evaluated the best subset selection (BSS) algorithm's predictive performance on the six datasets using 5-fold cross-validation, where 4 folds were for training and 1 for validation. The prediction error is defined as:

$$\text{error}_{\text{pred}} = \frac{1}{n} \sum_{i=1}^{n} (y_i - \hat{y}_i)^2. \tag{27}$$

The cross-validation score, averaged across the 5 folds, is shown in Figure 9. The enhanced algorithms demonstrate superior generalization and highlighting the advantage of the new criteria in selecting key features for predictive tasks.

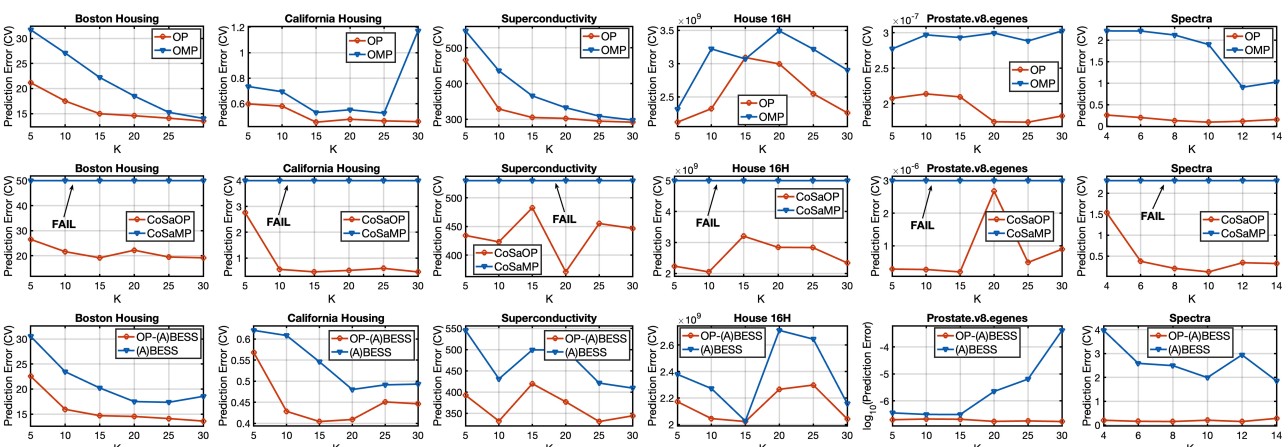

*Figure 9.* Rows 1–3 present the meta-gains in cross-validation performance in prediction ($\text{error}_{\text{pred}}$, smaller is better) for the Boston Housing, California Housing, Superconductivity datasets, House 16H, Prostate.v8.egenes, and Spectra datasets, respectively, across three algorithms as the number of selected features $K$ varies.

## S. The Optimal Gradient Pursuit Criteria

**The Optimal Gradient Pursuit Criteria:**

$$j^* = \text{argmax}_{j \in S^c} \frac{||\mathbf{X}_{S_{k-1}}^T \mathbf{r}^k||_2^2 + \left(\mathbf{r}^{k^T} \mathbf{X}_j\right)^2}{||\mathbf{X}_{S_{k-1}} \mathbf{X}_{S_{k-1}}^T \mathbf{r}^k + \mathbf{X}_j \mathbf{X}_j^T \mathbf{r}^k||_2}, \tag{28}$$

where **the underlined part only needs to be computed once** and its overall computational complexity is at the same magnitude as the correlation-based selection criterion in Gradient Pursuit. The idea of optimal gradient pursuit can be illustrated in Figure 10.

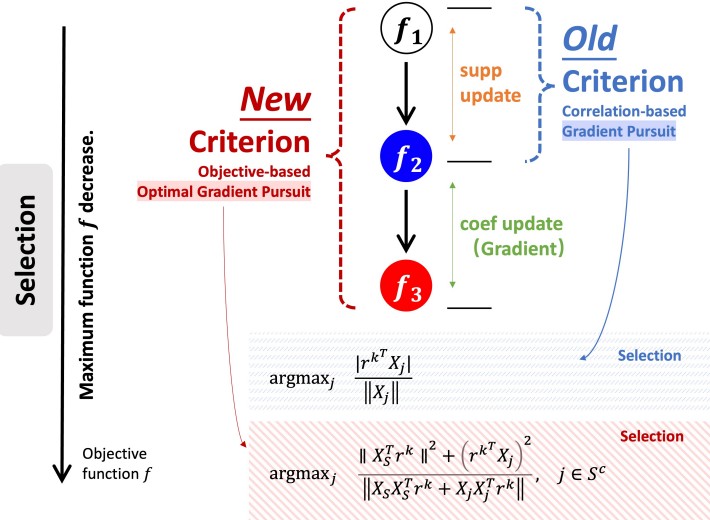

*Figure 10.* The idea of Optimal Gradient Pursuit.

***Remark*** S.1. The criterion (28) is the OGP version of the OP feature selection criterion (8). Similarly, by applying techniques analogous to those in the Appendix T, we can derive the OGP version of the feature elimination criterion (10), which we omit here for brevity.

*Remark* S.2. The criterion (28) does not repeatedly select features that have already been chosen. In practice, if adding a new feature at every iteration is not desired, one may consider using

$$
j^* = \operatorname{argmax}_j \begin{cases} \dfrac{\underbrace{||\mathbf{X}_{S_{k-1}}^T \mathbf{r}^k||_2^2} + \left(\mathbf{r}^{k^T}\mathbf{X}_j\right)^2}{\underbrace{||\mathbf{X}_{S_{k-1}}\mathbf{X}_{S_{k-1}}^T \mathbf{r}^k} + \mathbf{X}_j\mathbf{X}_j^T \mathbf{r}^k||_2}, & j \in S^c \\[4mm] \dfrac{\underbrace{||\mathbf{X}_{S_{k-1}}^T \mathbf{r}^k||_2^2}}{\underbrace{||\mathbf{X}_{S_{k-1}}\mathbf{X}_{S_{k-1}}^T \mathbf{r}^k||_2}} + \tau . & j \in S. \end{cases} \tag{29}
$$

Here, $\tau$ is a chosen threshold. In other words, when the effect of selecting a new feature is not sufficiently significant, the algorithm can instead continue performing gradient updates on the current support set $S$.

*Remark* S.3. Naturally, the algorithms discussed in the paper have corresponding OGP-accelerated versions.

*Remark* S.4. The conjugate gradient pursuit and other direction pursuit methods mentioned in (Blumensath & Davies, 2008) can also be updated using the idea of Optimal Pursuit.

## T. Derivation of Optimal Gradient Pursuit Criteria

*Proof.* Considering the algorithmic procedure of gradient pursuit in (Blumensath & Davies, 2008), it is necessary to take the effect of gradient updates into account, which leads to the derivation of the Optimal Gradient Pursuit criterion. The gradient on the subset $S = S_{k-1} \cup \{j\}$ is:

$$
\mathbf{g}_S = -\mathbf{X}_S^T \mathbf{r}^k. \tag{30}
$$

The exact line search step size along the gradient direction on the support set $S$ is given by:

$$
\alpha^k = \frac{\mathbf{g}_S^T \mathbf{g}_S}{\mathbf{g}_S^T \mathbf{X}_S^T \mathbf{X}_S \mathbf{g}_S}. \tag{31}
$$

Hence, considering the gradient update into account, we have:

$$
\begin{aligned}
\mathbf{r}^{k+1} &= \mathbf{y} - \mathbf{X}_S \left(\boldsymbol{\beta}^{k-1}\big|_S - \alpha^k \mathbf{g}_S\right) \\
&= \mathbf{r}^k + \alpha^k \mathbf{X}_S \mathbf{g}_S.
\end{aligned}
$$

And $\mathbf{r}^{k+1} \perp \mathbf{X}_S \mathbf{g}_S$ since

$$
\begin{aligned}
\mathbf{r}^{k+1^T}\mathbf{X}_S\mathbf{g}_S &= \mathbf{r}^{k^T}\mathbf{X}_S\mathbf{g}_S + \alpha^k \mathbf{g}_S^T\mathbf{X}_S^T\mathbf{X}_S\mathbf{g}_S \\
&= -\mathbf{g}_S^T\mathbf{g}_S + \alpha^k \mathbf{g}_S^T\mathbf{X}_S^T\mathbf{X}_S\mathbf{g}_S \\
&= 0.
\end{aligned}
$$

Hence, $||\mathbf{r}^k||_2^2 - ||\mathbf{r}^{k+1}||_2^2 = ||\alpha^k \mathbf{X}_S \mathbf{g}_S||_2^2$. The goal is to maximize the gradient update gains:

$$
\begin{aligned}
||\alpha^k \mathbf{X}_S\mathbf{g}_S||_2 &= \left\| \frac{\mathbf{g}_S^T\mathbf{g}_S}{\mathbf{g}_S^T\mathbf{X}_S^T\mathbf{X}_S\mathbf{g}_S} \mathbf{X}_S\mathbf{g}_S \right\|_2 \\
&= \left\| \frac{\mathbf{r}^{k^T}\mathbf{X}_S\mathbf{X}_S^T\mathbf{r}^k}{||\mathbf{X}_S\mathbf{X}_S^T\mathbf{r}^k||_2^2} \mathbf{X}_S\mathbf{X}_S^T\mathbf{r}^k \right\|_2 \\
&= \frac{\mathbf{r}^{k^T}\mathbf{X}_S\mathbf{X}_S^T\mathbf{r}^k}{||\mathbf{X}_S\mathbf{X}_S^T\mathbf{r}^k||_2} \\
&= \frac{\underbrace{||\mathbf{X}_{S_{k-1}}^T \mathbf{r}^k||_2^2} + \left(\mathbf{r}^{k^T}\mathbf{X}_j\right)^2}{\underbrace{||\mathbf{X}_{S_{k-1}}\mathbf{X}_{S_{k-1}}^T \mathbf{r}^k} + \mathbf{X}_j\mathbf{X}_j^T \mathbf{r}^k||_2}.
\end{aligned}
$$

The underlined part only needs to be computed once. The proof is complete. $\qquad \square$

## U. Theoretical Results and Experiments of Optimal Gradient Pursuit

The algorithmic procedure of feature selection algorithm guided by criterion (28): Optimal Gradient Pursuit, follows the similar structure as gradient pursuit in (Blumensath & Davies, 2008), with the selection criterion in gradient pursuit being meta-substituted by criterion (28). We first show that Optimal Gradient Pursuit possesses similarly strong theoretical properties as Gradient Pursuit. The following theorem corresponds to Theorem 3 in (Blumensath & Davies, 2008).

**Theorem U.1.** *There exists a constant $c < 1$, which only depends on* $\mathbf{X}$*, such that the residual calculated with Optimal Gradient Pursuit decays as*

$$\left\|\mathbf{r}^k\right\|_2^2 \leq c \left\|\mathbf{r}^{k-1}\right\|_2^2. \tag{32}$$

*Proof.* For the same $\mathbf{r}^{k-1}$, the residual at step $k$ of OGP and GP satisfies:

$$\left\|\mathbf{r}_{\mathrm{OGP}}^k\right\|_2^2 \leq \left\|\mathbf{r}_{\mathrm{GP}}^k\right\|_2^2$$

by the derivation of OGP criterion (28) in Appendix T. By Theorem 3 in (Blumensath & Davies, 2008),

$$\left\|\mathbf{r}_{\mathrm{GP}}^k\right\|_2^2 \leq c \left\|\mathbf{r}^{k-1}\right\|_2^2.$$

The derivation of this theorem is independent of the historical iteration steps. Hence for Optimal Gradient Pursuit,

$$\left\|\mathbf{r}^k\right\|_2^2 \leq c \left\|\mathbf{r}^{k-1}\right\|_2^2.$$

The proof is complete. □

**Experiment on Residual Convergence:**

Experimental result on residual convergence of Optimal Gradient Pursuit and Gradient Pursuit is shown in Figure 11. Optimal Gradient Pursuit exhibits better residual convergence compared to Gradient Pursuit on different test cases. This is also consistent with the results presented in the Appendix T and Theorem U.1.

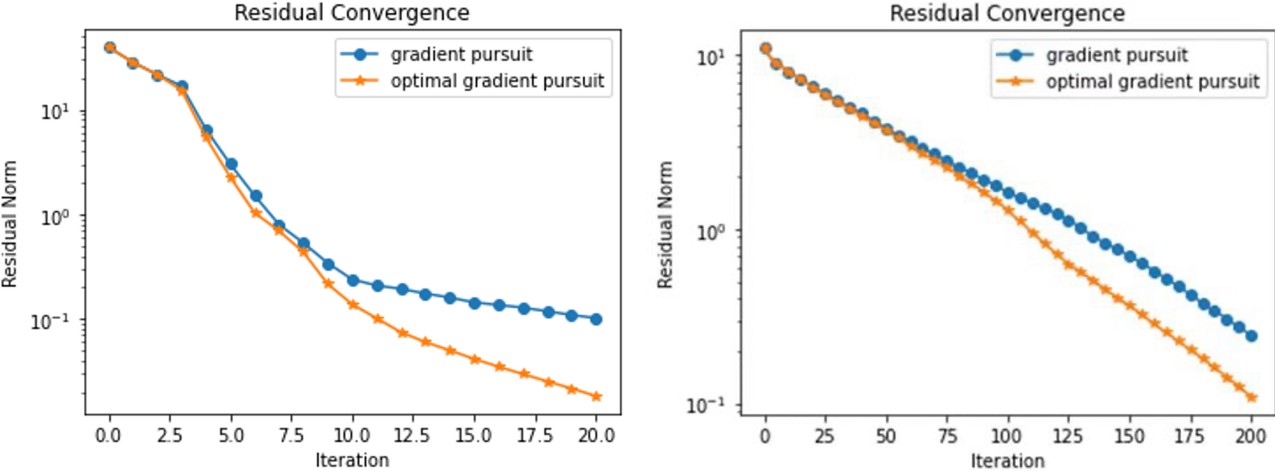

*Figure 11.* Residual convergence of Optimal Gradient Pursuit and Gradient Pursuit.

**Experiment on Computation Time:**

We further compared the runtime of GP and OGP, as shown in Figure 12.

Both methods achieve an order-of-magnitude speedup compared to the least squares-based subset selection approach. For ultra-high-dimensional features, OGP provides an efficient acceleration scheme for the optimal pursuit strategy. Specifically, since gradients can be easily computed for general functions, the OGP method can be extended to general objective functions, offering a promising direction for further expanding the applicability of optimal pursuit in future research.

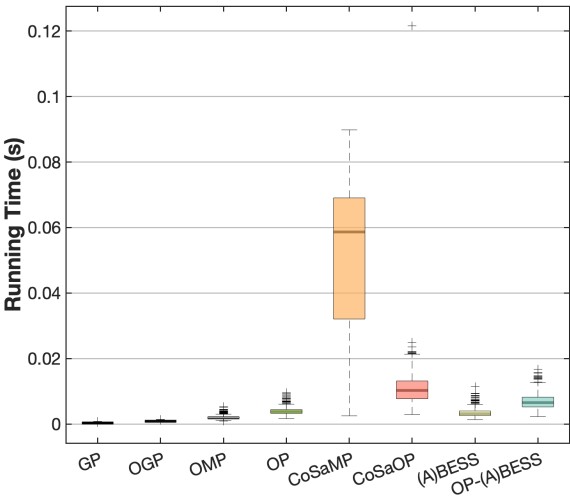

*Figure 12.* Running time for each algorithm over 500 independent runs.

## V. Optimal Pursuit for Column Subset Selection

**Optimal Pursuit Criteria for Column Subset Selection:**

We can follow the procedure in Section 3 to construct the feature selection and elimination subproblems for column subset selection. By solving them using approaches as those in Appendices A–C, we obtain the Optimal Pursuit feature selection and elimination criteria for the column subset selection problem as follows:

$$\textbf{Selection:} \quad \underset{j \in S_{k-1}^c}{\arg\max} \frac{||\mathbf{R}^{k^T}\mathbf{X}_j||^2}{\mathbf{X}_j^T \left(\mathbf{I} - \mathbf{X}_{S_{k-1}}\left(\mathbf{X}_{S_{k-1}}^T \mathbf{X}_{S_{k-1}}\right)^{-1}\mathbf{X}_{S_{k-1}}^T\right)\mathbf{X}_j}.$$

$$\textbf{Elimination:} \quad \underset{j \in S_{k-1}}{\arg\max} \ \text{trace}\left(\mathbf{X}^T\mathbf{X}_{S_{k-1}}\left(\mathbf{I} - \mathbf{e}_j\mathbf{e}_j^T\right)\left(\mathbf{C}_{k-1} - \frac{\mathbf{C}_{k-1}\mathbf{e}_j\mathbf{e}_j^T\mathbf{C}_{k-1}}{\mathbf{e}_j^T\mathbf{C}_{k-1}\mathbf{e}_j}\right)\left(\mathbf{I} - \mathbf{e}_j\mathbf{e}_j^T\right)\mathbf{X}_{S_{k-1}}^T\mathbf{X}\right).$$

where $\mathbf{R}^k = \mathbf{X} - \mathbf{X}_{S_{k-1}}\mathbf{B}^{k-1}$ serves as the current residual. The definitions of $\mathbf{e}_j$ and $\mathbf{C}_{k-1}$ are the same as those in Theorem 3.11.

**Classic Criteria for Column Subset Selection:**

The classic criteria can also be similarly generalized to column subset selection problem by constructing subproblems as (P0) and (Q0) in Section 2. Specifically:

$$\textbf{Selection:} \quad \underset{j \in S_{k-1}^c}{\arg\max} \frac{||\mathbf{R}^{k^T}\mathbf{X}_j||^2}{\mathbf{X}_j^T\mathbf{X}_j}.$$

$$\textbf{Elimination:} \quad \underset{j \in S_{k-1}}{\arg\min} \ ||\mathbf{X}_j||_2^2 \cdot ||\mathbf{B}^{k-1}[i,:]||_2^2.$$

where $\mathbf{B}^{k-1}[i,:]$ is the $i$-th row of $\mathbf{B}^{k-1}$, and $j$ (in elimination) represents the $i$-th element of $S_{k-1}$ for $i = 1, 2, \ldots, |S_{k-1}|$.

*Remark* V.1. It is important to note that the results discussed in the main text remain valid for the criteria on column subset selection problem. **The computational complexity of the Optimal Pursuit feature selection and elimination criteria is of the same order as that of the classical criteria.**

**Experiment on Column Subset Selection:**

The experiment setting is:

- **Datasets:** We conduct experiments on eight standard grayscale 256×256 image datasets compiled from two sources [4].

- **Baselines:** We select baseline algorithm for CSS - the leverage score method. This approach relies on the right singular vectors from SVD decomposition; the more singular vectors used, the better the selection performance, but at the cost of more redundant SVD computations.

- **Evaluation Metrics:** We use $\|\mathbf{X} - \mathbf{X}_S\mathbf{B}\|_F/\|\mathbf{X}\|_F$ as the evaluation metrics.

The experimental results is shown in Table 3. In the table, SVD-128/256 refers to CSS based on leverage scores computed using 128/256 singular vectors. In the last column of the table, we provide the bound given by the optimal rank-$K$ approximation from SVD.

*Table 3.* Column Subset Selection. $\|\mathbf{X} - \mathbf{X}_S\mathbf{B}\|_F/\|\mathbf{X}\|_F$ are shown below, with the best in **bold** and the second-best underlined. SVD-128/256 refers to CSS based on leverage scores computed using 128/256 singular vectors (the more singular vectors used, the better the selection performance, but at the cost of more redundant SVD computations.). In the last column of the table, we provide the bound given by the optimal rank-K approximation from SVD.

| Dataset | $K$ | OMP | OP | CoSaMP | CoSaOP | (A)BESS | OP-(A)BESS | Leverage Score (SVD-128) | Leverage Score (SVD-256) | Bound by SVD |
|---|---|---|---|---|---|---|---|---|---|---|
| Mornach | 3 | 0.3881 | 0.3794 | 0.3945 | 0.3979 | 0.3884 | **0.3743** | 0.4262 | 0.4400 | 0.3330 |
| | 5 | 0.3521 | 0.3422 | 0.4193 | 0.3820 | 0.3662 | **0.3399** | 0.3966 | 0.3814 | 0.2971 |
| | 10 | 0.2953 | 0.2758 | 0.4246 | 0.3275 | 0.3436 | **0.2771** | 0.3459 | 0.3079 | 0.2268 |
| | 15 | 0.2466 | 0.2330 | 0.3951 | 0.2468 | 0.3285 | **0.2318** | 0.2845 | 0.2770 | 0.1804 |
| | 20 | 0.2074 | 0.1999 | 0.3781 | 0.2150 | 0.2769 | **0.1981** | 0.2648 | 0.2419 | 0.1495 |
| Barbara | 3 | 0.2823 | 0.2520 | 0.3256 | 0.2571 | 0.2831 | **0.2505** | 0.3284 | 0.2981 | 0.2296 |
| | 5 | 0.2403 | **0.2173** | 0.3331 | 0.2303 | 0.2560 | 0.2179 | 0.3146 | 0.2610 | 0.1882 |
| | 10 | 0.1831 | 0.1743 | 0.3152 | 0.1896 | 0.2437 | **0.1730** | 0.2622 | 0.2197 | 0.1431 |
| | 15 | 0.1577 | **0.1458** | 0.3023 | 0.1576 | 0.2306 | 0.1468 | 0.2321 | 0.1847 | 0.1147 |
| | 20 | 0.1337 | 0.1273 | 0.2871 | 0.1405 | 0.2242 | **0.1263** | 0.2156 | 0.1585 | 0.0983 |
| Boats | 3 | 0.2589 | 0.2453 | 0.3333 | 0.2589 | 0.2741 | **0.2435** | 0.3139 | 0.2995 | 0.2077 |
| | 5 | 0.2219 | 0.2062 | 0.3149 | 0.2176 | 0.2676 | **0.2045** | 0.2725 | 0.2640 | 0.1688 |
| | 10 | 0.1602 | **0.1575** | 0.2961 | 0.1747 | 0.2327 | 0.1578 | 0.2159 | 0.2004 | 0.1293 |
| | 15 | 0.1336 | 0.1330 | 0.2779 | 0.1527 | 0.2187 | **0.1311** | 0.2015 | 0.1625 | 0.1050 |
| | 20 | 0.1148 | 0.1148 | 0.2460 | 0.1262 | 0.1549 | **0.1131** | 0.1810 | 0.1436 | 0.0878 |
| House | 3 | 0.1983 | **0.1940** | 0.2658 | 0.2072 | 0.2097 | **0.1940** | 0.2418 | 0.2014 | 0.1677 |
| | 5 | 0.1671 | 0.1618 | 0.2581 | 0.1988 | 0.2020 | **0.1617** | 0.2311 | 0.1835 | 0.1378 |
| | 10 | 0.1286 | 0.1219 | 0.2564 | 0.1386 | 0.1878 | **0.1195** | 0.1927 | 0.1552 | 0.0977 |
| | 15 | 0.1019 | 0.0969 | 0.2067 | 0.1086 | 0.1793 | **0.0963** | 0.1502 | 0.1331 | 0.0762 |
| | 20 | 0.0785 | 0.0785 | 0.2331 | 0.0968 | 0.1740 | **0.0774** | 0.1332 | 0.1158 | 0.0606 |
| Lena | 3 | 0.2629 | **0.2572** | 0.2936 | 0.2951 | 0.2675 | 0.2575 | 0.3748 | 0.3722 | 0.2368 |
| | 5 | 0.2328 | **0.2234** | 0.2835 | 0.2430 | 0.2616 | 0.2255 | 0.3511 | 0.2977 | 0.1987 |
| | 10 | 0.1830 | 0.1731 | 0.2592 | 0.1925 | 0.2313 | **0.1695** | 0.3362 | 0.2330 | 0.1414 |
| | 15 | 0.1457 | **0.1445** | 0.2430 | 0.1496 | 0.1821 | 0.1592 | 0.3147 | 0.1795 | 0.1170 |
| | 20 | 0.1301 | 0.1257 | 0.2306 | 0.1321 | 0.1867 | **0.1256** | 0.3072 | 0.1710 | 0.0986 |
| Parrot | 3 | 0.2460 | **0.2383** | 0.2953 | 0.2634 | 0.2524 | 0.2411 | 0.4502 | 0.3177 | 0.2000 |
| | 5 | 0.2109 | 0.2049 | 0.2914 | 0.1978 | 0.2498 | **0.1965** | 0.3146 | 0.2546 | 0.1660 |
| | 10 | 0.1598 | 0.1547 | 0.2645 | 0.1609 | 0.2452 | **0.1502** | 0.2682 | 0.2163 | 0.1237 |
| | 15 | 0.1377 | **0.1239** | 0.2551 | 0.1362 | 0.2334 | **0.1239** | 0.2606 | 0.1657 | 0.0980 |
| | 20 | 0.1163 | **0.1054** | 0.2485 | 0.1239 | 0.2213 | 0.1062 | 0.2186 | 0.1507 | 0.0812 |
| Cameraman | 3 | 0.2749 | 0.2767 | 0.3727 | 0.2825 | 0.3041 | **0.2747** | 0.4385 | 0.5908 | 0.2433 |
| | 5 | 0.2456 | **0.2443** | 0.3684 | 0.2567 | 0.3019 | 0.2449 | 0.3730 | 0.3494 | 0.2091 |
| | 10 | 0.2017 | 0.1953 | 0.3596 | 0.2282 | 0.2958 | **0.1945** | 0.3298 | 0.2348 | 0.1639 |
| | 15 | 0.1729 | 0.1691 | 0.3527 | 0.1799 | 0.1835 | **0.1680** | 0.3035 | 0.2231 | 0.1347 |
| | 20 | 0.1548 | 0.1488 | 0.3449 | 0.1732 | 0.1805 | **0.1480** | 0.2922 | 0.1966 | 0.1143 |
| Foreman | 3 | 0.1923 | 0.1911 | 0.2751 | 0.2154 | 0.2203 | **0.1892** | 0.2870 | 0.2363 | 0.1465 |
| | 5 | 0.1473 | 0.1457 | 0.2675 | 0.2183 | 0.2190 | **0.1438** | 0.2578 | 0.1683 | 0.1154 |
| | 10 | 0.1060 | 0.1000 | 0.2564 | 0.1140 | 0.2059 | **0.1017** | 0.2273 | 0.1419 | 0.0771 |
| | 15 | **0.0792** | 0.0805 | 0.2484 | 0.1009 | 0.1966 | **0.0792** | 0.2221 | 0.1293 | 0.0593 |
| | 20 | 0.0669 | 0.0667 | 0.2463 | 0.1025 | 0.1874 | **0.0658** | 0.2175 | 0.0866 | 0.0501 |

[4]Available at http://dsp.rice.edu/software/DAMP-toolbox and http://see.xidian.edu.cn/faculty/wsdong/NLR_Exps.htm

The experimental results show that the enhanced algorithm achieves a significantly better performance than the original algorithm in the CSS task. **The original algorithm generally outperforms the leverage score methods (SVD-128), while the enhanced algorithm consistently surpasses the leverage score method (SVD-256).**

Notably, **OP-(A)BESS achieves SOTA performance on this task, with approximation errors approaching the optimal bound given by SVD.** This further highlights the substantial advantage of the new criteria on the CSS task.

## W. Complex Signal Processing

In this task, we tested a 128-dimensional complex signal with 20 frequency components, applying the BSS algorithm to estimate the frequency components on an $2\times$ oversampled Fourier domain. We used two evaluation metrics: cosine similarity (Corr) to assess the accuracy of amplitude recovery (higher is better) and Complementary Cumulative Distribution Function (CCDF) to measure frequency estimation error probability (lower is better).

By the high correlation between nearby frequency components, this also constitutes a test scenario with highly correlated features. To evaluate algorithm performance, the target signal is constructed with closely spaced frequency components, resulting in high correlation among features in the true subset. The frequency domain visualization, radar plot, and the performance of these two metrics are shown in Figure 13.

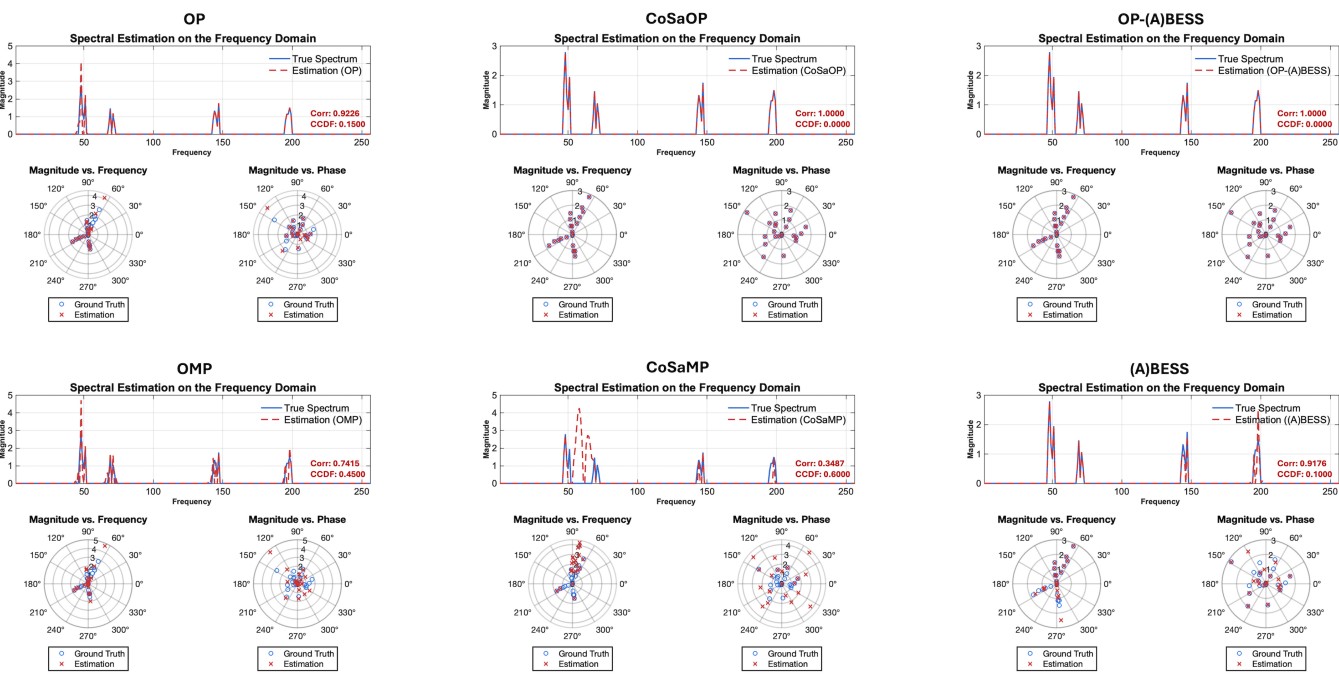

*Figure 13.* Comparison of Optimal Pursuit enhanced algorithms and classic algorithms on complex signal processing.

From the experimental results, it is evident that the enhanced algorithm achieves significantly lower frequency estimation error probability and higher correlation in the line spectrum estimation task. Notably, OP-(A)BESS and CoSaOP achieve perfect estimation in this task.

Since features in the line spectrum estimation problem are also highly correlated, these results further validate the substantial advantage of our proposed criteria in scenarios with high feature correlation.

