# OpenReview forum: "Best Subset Selection: Optimal Pursuit for Feature Selection and Elimination"
_ICML.cc/2025/Conference — ICML 2025 poster_

### Official Review · Reviewer_g7tW · 2025-02-26

**Overall Recommendation:** 2

**Summary:**

This paper introduces optimal pursuit strategies for feature selection and elimination in best subset selection problems. It challenges classical feature selection methods by offering new selection and elimination criteria, which focus on feature interactions as opposed to individual significance alone. The authors revisit the classic greedy algorithms, such as Matching Pursuit and Orthogonal Matching Pursuit, and propose enhanced algorithms by substituting their classical feature importance criteria with the new optimal pursuit criteria. The results demonstrate that these new methods outperform traditional approaches.

**Claims And Evidence:**

Please refer to **Questions For Authors**.

**Essential References Not Discussed:**

N/A

**Experimental Designs Or Analyses:**

Please refer to **Questions For Authors**.

**Methods And Evaluation Criteria:**

Please refer to **Questions For Authors**.

**Other Comments Or Suggestions:**

Please refer to **Questions For Authors**.

**Other Strengths And Weaknesses:**

**Strengths:**

1. The paper challenges existing greedy algorithms by proposing a more holistic method to evaluate feature importance, accounting for interactions between features, leading to more optimal feature selection.

2. The paper is mathematically rigorous and builds a strong theoretical foundation for the proposed optimal selection and elimination criteria.

3. The algorithms are experimentally validated across tasks like compressed sensing and sparse regression, showing clear improvements in performance metrics such as recovery rates and computational time efficiency.

4. Despite the added complexity of the new criteria, the algorithms maintain the computational efficiency of the classical greedy methods.

**Weaknesses:**

Please refer to **Questions For Authors**.

**Questions For Authors:**

**Weaknesses:**
﻿
1. The proposed optimal pursuit strategy involves solving optimization subproblems for feature selection and elimination in a more detailed manner, incorporating interactions between features. While mathematically elegant, this additional complexity can make the algorithms harder to implement and computationally expensive for practitioners. The algorithms also involve matrix inversions, which could potentially increase the time complexity and make them unsuitable for high-dimensional problems where feature selection is crucial.
﻿
2. Although the paper demonstrates the effectiveness of the proposed algorithms in compressed sensing and sparse regression tasks, these are relatively specific domains. The algorithms are not evaluated on a wider variety of machine learning tasks or diverse datasets.
﻿
3. While the authors present important optimizations to classical feature selection algorithms, the core contribution seems to build upon existing algorithms. Many of the proposed algorithms can be seen as modifications of existing approaches rather than a completely novel class of algorithms. This lack of a breakthrough contribution might limit the paper’s impact and novelty compared to other works in the field.
﻿
4. While the paper is mathematically thorough, it lacks detailed practical insights or step-by-step implementation guidelines for those who might want to apply these new algorithms. There is minimal discussion on potential limitations or challenges in applying these methods in real-world machine learning systems.
﻿
5. The paper uses standard metrics like NMSE and $R^2$ for evaluation, which are common in compressed sensing and sparse regression tasks. However, the paper does not explore a wider variety of evaluation criteria, such as cross-validation performance or other task-specific metrics.

﻿**Conclusion:**

The weaknesses primarily stem from the increased complexity and limited generalization of the proposed methods, along with an incremental contribution that builds upon existing techniques. While the proposed algorithms are a meaningful enhancement for specific tasks like compressed sensing, they may not significantly advance the field of feature selection in a broad sense. Furthermore, the lack of a deep dive into practical implementation challenges and a broader experimental validation means that the paper may not be as impactful or widely applicable in its current form. I hope the author can address my concerns and change my expected rating.

**Relation To Broader Scientific Literature:**

Please refer to **Other Strengths And Weaknesses**.

**Theoretical Claims:**

Please refer to **Questions For Authors**.

---

> ### Author Rebuttal · Authors · 2025-04-01
>
> We sincerely appreciate your feedback and constructive suggestions on our paper, which will help enrich the original content. In this rebuttal, we address the concerns raised in the reviews. For references [1-8] in the rebuttal, please refer to Reviewer uuYf.
>
> **Q1: (Complexity)**  Thank you for your excellent question, which has driven us to further our advancements.
>
> First, the algorithmic complexity of our proposed criteria-enhanced algorithm remains at the same order as the original methods. Both rely on solving the least squares problem over a given subset $S$, which involves a linear system solved via Cholesky decomposition. As noted in Remark 3.7 and 3.15 of our paper, given the Cholesky decomposition, computing the matrix inverse requires only $O(K^2)$ complexity. Thus, despite the presence of inverse terms in our new metric, its computational complexity remains fundamentally equivalent to that of the original algorithm.
>
> However, in ultra-high-dimensional settings, solving least squares over a subset can be prohibitive, affecting both the original and enhanced algorithms. Updating coefficients via least squares is equivalent to Newton’s method. [4] proposed Gradient Pursuit, which follows the same correlation-based selection strategy but replaces Newton’s method with gradient-based updates, significantly reducing computational overhead.
>
> Our optimal pursuit idea extends to Gradient Pursuit, forming Optimal Gradient Pursuit (OGP), which simultaneously considers support set updates and coefficient updates while maintaining the same computational complexity as Gradient Pursuit.
>
> In this rebuttal, we explicitly derive the selection criterion for OGP:
> \begin{equation}\arg\max_{j} \begin{cases}
> \frac{\underline{||X_S^Tr^k||^2}+({r^k}^TX_j)^2}{||\underline{X_SX_S^Tr^k} +X_jX_j^Tr^k||}, & j \in S^c \\\\
> \frac{\underline{||X_S^Tr^k||^2}}{||\underline{X_SX_S^Tr^k}||}, & j \in S
> \end{cases}\end{equation}
> where the underlined part only needs to be computed once, keeping the overall complexity comparable to correlation-based selection in Gradient Pursuit. We establish OGP’s convergence theory and validate it with numerical experiments, demonstrating superior performance both theoretically and empirically.
> [FigOGP](https://drive.google.com/file/d/1kx7exKHUYToPS3wYuG4yTsVowCtkv92e/view?usp=sharing)
>
> We compared GP and OGP runtime on numerical examples from our paper:
> [FigTime](https://drive.google.com/file/d/1RJ8yreuaGvReU2svoT-OrXCY4ILEZlMv/view?usp=sharing)
>
> Both methods achieve an order-of-magnitude speedup over least squares-based subset selection. OGP provides an efficient acceleration scheme for the optimal pursuit strategy, extending its applicability to general objective functions in future research.
>
> **Q2 (Diverse datasets, machine learning tasks, and metrics evaluation)**  Thank you for your suggestion. We have conducted additional tests across five tasks, ten more datasets, and six metrics. Due to space constraints, please refer to Reviewer sDws (Q2) for details.
>
> **Q3 (Contributions)**  Thanks for your question. As stated in [2] (Section 3.3.2, pp. 59–60), best subset selection is built on forward feature selection and backward elimination. Greedy algorithms combine these criteria, with classical criteria based on correlation and T-statistics.
>
> Our contribution lies in re-examining these foundational criteria from an optimization perspective. By modeling feature significance and interaction through a block coordinate descent framework, we clarified the optimization essence of classical criteria and proposed new selection and elimination models. Using forward and backward matrix inversion techniques, we derived explicit new criteria, providing a foundation for future best subset selection algorithms.
>
> Additionally, we further analyzed:
> (1) Theoretical behavior under high feature correlations (Reviewer aZmj, Theorem 1\&2).
> (2) Complexity and algorithmic convergence.
> (3) Empirical performance gains.
> (4) Performance across various machine learning tasks and metrics.
>
> Our work has significant potential:
>
> (1) Theoretically, our convergence results suggest the possibility of breaking existing RIP assumptions, advancing algorithmic study in this NP-hard problem.
> (2) Practically, our new criteria significantly improve performance across machine learning tasks, datasets, and evaluation metrics, laying a foundation for future algorithm design.
>
> Furthermore, our optimal pursuit idea extends to other greedy methods, such as Optimal Gradient Pursuit.
>
> **Q4 (Implementation)** Thank you for your suggestion. We recognize the importance of practical implementation and plan to open-source all code, covering the original algorithm, extensions, and accelerated optimal gradient pursuit, along with detailed workflow guidance.
>
> We will also provide tutorials on applying the algorithm to best subset selection, column subset selection, line spectrum estimation, and other machine learning applications.

---

### Official Review · Reviewer_aZmJ · 2025-03-10

**Overall Recommendation:** 2

**Summary:**

This paper proposes two criteria for feature selection and feature elimination in the context of solving the best subset selection problem. The authors approach these criteria from an optimization perspective. These criteria can be incorporated into various heuristic subset selection algorithms. Additionally, the authors establish convergence guarantees for one such algorithm, CoSaOP. Numerical experiments are reported to demonstrate the effectiveness of new criteria against classical feature selection and elimination approaches. In particular, Sections 2 and 3 develop an optimization-based framework for feature selection and elimination. The authors highlight the limitations of classical criteria, which partially capture variations in the objective function due to feature addition or removal, and subsequently propose refined criteria to address these shortcomings. Furthermore, the two sub-problems are reformulated to enhance computational efficiency. Section 4 provides convergence guarantees for the CoSaOP algorithm. Section 5 presents numerical experiments on synthetic and real-world datasets, demonstrating the practical advantages of the proposed approach over existing heuristic methods.

**Claims And Evidence:**

The claims made in the submission are clear and also supported by their numerical experiments.

**Essential References Not Discussed:**

NA

**Experimental Designs Or Analyses:**

Experimental designs are sound.

The experimental setup for the synthetic datasets would benefit from additional details. In Section 5.1.1, the generation procedures for the sparse vector $\beta$ and the random Gaussian matrix $X$ are not clearly specified. Furthermore, the definition of the signal-to-noise ratio (SNR) is absent from the main text. To enhance the rigor of the study, the authors are encouraged to conduct further experiments with higher-dimensional settings and under conditions where the SNR values are low.

**Methods And Evaluation Criteria:**

The proposed methods, i.e., CoSaOP, partially make sense under the assumptions provided in Appendix F.

The benchmark instances used in Section 5.1.1 are okay. However, the instances used in Sections 5.1.2 & 5.2 might not ensure the assumptions provided in Appendix F.

**Other Comments Or Suggestions:**

Here are some minor comments for the authors to consider:

1. Following the introduction, it would be beneficial to include a notation convention section to clarify the terminology used throughout the main text.

2. In Remark 3.7, the authors assert that the proposed criteria do not introduce significant additional computational costs. An algorithmic complexity analysis should be provided to substantiate this claim.

**Other Strengths And Weaknesses:**

Weakness:

The paper is well-organized. However, the clarity of writing could be improved, and some key assumptions are omitted from the main content, which may significantly impact or weaken the theoretical results.

The originality is incremental. The proposed selection and elimination method follows a similar idea from vanilla local search, where the term "optimal" is based on maximizing over all possible single selection/elimination index. It is better to demonstrate whether the convergence results for the proposed optimal criteria would also hold for other greedy criteria.

**Questions For Authors:**

NA

**Relation To Broader Scientific Literature:**

NA

**Theoretical Claims:**

The proofs for theoretical results look correct on my side. However, the theoretical results provided in Section 4 need further assumptions and conditions, which may weaken the main contributions.

---

> ### Author Rebuttal · Authors · 2025-04-01
>
> We sincerely appreciate your insightful question, which has driven us to further theoretical advancements. In this rebuttal, we address concerns raised in the reviews. For references [1-8], please refer to Reviewer uuYf.
>
> **Q1 (Theoretical Assumptions):**
> Thank you for your question. The theoretical assumptions about algorithmic convergence in our work align with Section 2.3 of the CoSaMP paper [1]. Other best subset selection algorithms also rely on assumptions like the Restricted Isometry Property (RIP) due to the NP-hard nature of the problem. Without such assumptions, proving convergence for any polynomial-time algorithm would be intractable unless P = NP.
>
> However, your question led us to reflect further. Empirical observations reveal that even when theoretical assumptions are violated (e.g., correlated features), our proposed algorithm family performs well. This stems from our criteria’s explicit focus on feature interaction, motivating new theoretical frameworks under violated RIP conditions.
>
> **Theorem 1** Suppose the true subset $S^*$ contains indices $(i, j)$, where feature correlation is $\rho$. Assuming $S$ includes $i$, then for the classical criterion:
> \begin{equation}
>   \frac{|{r^k}^T X_j|}{||X_j||_2} \le \sqrt{1-\rho^2}||r^k||_2,\tag{C1}
> \end{equation}
> while our objective-based criterion (8) satisfies
> \begin{equation}
>   \frac{({r^k}^T X_j)^2}{X_j^T(I-X_S(X_S^TX_S)^{-1}X_S^T)X_j} \ge \frac{1}{1-\rho^2}\left(\frac{{r^k}^TX_j}{||X_j||_2}\right)^2.\tag{C2}
> \end{equation}
>
> Theorem 1 shows that under strong feature correlation, traditional criteria struggle to identify true features, while our criterion (8) provides a stable lower bound, mitigating correlation effects. If equality holds in (C1), substituting into (C2) eliminates dependence on $\rho$, fully removing correlation influence. Additionally, (C2) connects our criterion (8) to classical ones.
>
> Pseudo-correlation may arise in best subset selection. Pseudo-correlated features $X_p$ are highly correlated with important features $X_i$ in $S^*$ but do not belong to $S^*$. Classical T-statistics-based criteria struggle to remove such features, while our criterion (10) reliably identifies them.
>
> **Theorem 2**
> 1) In noiseless cases, if $S^* \subset S$, then for all $j_m \in S \setminus S^*$,
> \begin{equation*}
> j_m \in \arg\max_{j \in S}~\text{objective-based criterion (10)},
> \end{equation*}
> whereas classical criterion (4) lacks this guarantee.
> 2) If $X_p \in S$ is pseudo-correlated with $X_i \in S^*$ (correlation $1 - \epsilon$), when $\epsilon$ is small, classical criteria (4) may discard true features, while our criterion (10) correctly removes $X_p$.
>
> These theorems show our criterion (8) effectively identifies key features under strong correlation, and criterion (10) removes pseudo-correlated features. Experimental results in Q2 confirm superior performance. Theorems 1 and 2 pave the way for research on algorithm convergence under weakened RIP conditions. The detailed proof is provided in [The detailed proof](https://drive.google.com/file/d/1r7Y-2DZ07TD7ORxU72hwJi73dOFtIo29/view?usp=sharing)
>
> **Q2 (Experimental Design)**
> Thank you for your suggestions. Due to character limits, we will include details on sparse vector and random Gaussian matrix generation, along with SNR definitions, in the revised paper. Additional comparisons were conducted under extreme conditions:
>
> 1. Small sample rate, high-dimensional vectors: $p = 2000$, with $n/p$ varying from 0.05 to 0.1.
> 2. High noise: SNR from 5 to 15.
> 3. Highly correlated features (RIP violated): The covariance matrix of $X$ follows a Toeplitz structure, where $\text{corr}_{ij} = \rho^{|i-j|}$ with $\rho = 0.7$.
>
> Sparse vectors with sparsity level $K = 10$ were used. Phase transition diagrams illustrate how varying sampling rates and SNR impact performance. Larger blue areas indicate stronger performance.
> [Phase Transition](https://drive.google.com/file/d/1HY6-6XzeVTq-LUtehvQS1hZgjVvNSu7k/view)
>
> Results show that all algorithms enhanced with our criteria exhibit major improvements in phase transition capabilities, validating both theoretical insights from Q1 and advantages in high-dimensional, low-SNR cases.
>
> Additional tests included sparse regression datasets, cross-validation, column subset selection, line spectral estimation, and other machine learning problems where feature correlation is prevalent. Enhanced algorithms consistently outperformed others.
>
> **Q3 (RHS of Eq.(19))**:
> Thank you for your question. The matrix 2-norm used is: $\text{norm}(D,2)= \\sqrt{\\lambda_{\\max}(D^TD)}.$
>
> Since $D$ is diagonal, $\text{norm}(D,2) = \\max |D_{jj}|$. This justifies why Eq.(19) holds.
>
> **Q4 (Contributions and Other Greedy Criteria)**
> Thank you for your question. Due to space constraints: For the contribution of this paper, see Reviewer g7tW (Q3). For other greedy criteria, we extend the optimal pursuit idea to Gradient Pursuit (Reviewer g7tW, Q1) and Column Subset Selection (Reviewer aDws, Q2, 4).

---

> > ### Comment · Reviewer_aZmJ · 2025-04-08
> >
> > Thank you for the response. While I found clarification helpful, I have to maintain my score based on the following concerns.
> >
> > **Theoretical Assumptions.**
> >
> > I appreciate the effort in providing additional theoretical results (Theorem 1 and Theorem 2). However, Theorem 1 only shows that the proposed criteria are better than the classical one, which meets our expectations due to a finer selecting \& removing step with a greater computational complexity. Additionally, the dependency on $\rho$ works for any choices within interval $(0,1)$, I cannot see something like "phase-transition" for strong correlation case.
> >
> > For Theorem 2, when $\epsilon$ is small, it is better to compare with existing theoretical/statistical results in (robust/perturbed) sparse regression, which ensures similar guarantees or bounds.
> >
> > **Experimental Design.**
> >
> > Usually, for high noise or low SNR setting, we set SNR < 1.
> >
> > Toeplitz structure is commonly used for input sample generation. However, the correlation between different features are highly-inconsistent, I do not think the resulting instances satisfies the assumed high-correlated condition/assumption.

---

> > > ### Author Response · Authors · 2025-04-08
> > >
> > > We sincerely appreciate your response on our rebuttal. Below, we address your concerns and provide further clarifications:
> > >
> > > **Q1 (Theorem 1):** Thanks for your question. Theorem 1 demonstrates that if features $i$ and $j$ in the true subset are highly correlated (larger $\rho$ means higher correlation), inequality (C1) implies that as their correlation $\rho$ approaches 1, feature $j$ becomes increasingly unidentifiable under traditional correlation-based criterion once feature $i$ has been selected, since the coefficient term $\sqrt{1-\rho^2}$ in the RHS converges to 0. In contrast, (C2) shows that when $\rho$ is large, since the first term’s coefficient in RHS $1/(1-\rho^2)$ is large (approaches $+\infty$ as $\rho$ approaches 1), the proposed criterion (8) exists a stable lower bound. **The larger the correlation $\rho$, the stronger discrepancy between traditional correlation-based criterion and proposed criterion.** This significantly mitigates the impact of feature correlations on identifying $j$, making true features more reliably detectable.
> > >
> > > We would like to clarify that the influence of feature correlations cannot be entirely eliminated—doing so would reduce the problem to RIP scenario, which would imply a solution of the NP-hard problem BSS in polynomial time. This is clearly unrealistic. The contribution of our proposed criteria lies in minimizing the impact of correlations through minimal modifications, as demonstrated above.
> > >
> > > **Q2 ((Robust/perturbed) sparse regression):** Thank you for your question. To the best of our knowledge, (robust/perturbed) sparse regression is primarily implemented through the following approaches:
> > > 1. **Adopting more robust loss functions**, such as the Huber loss.
> > > 2. **Introducing perturbation variables** and imposing robust regularization on the objective function, for example, via a total least squares objective.
> > > 3. **Resampling or perturbing data** to fit the model multiple times and selecting features that appear most frequently.
> > >
> > > However, all these methods necessitate **modification of the objective function (model)**. In other words, to achieve other goals, the target problem **no longer aligns with the Best Subset Selection (BSS) problem (2)** discussed in our work.
> > >
> > > **Theorem 2 in our study focuses specifically on the NP-hard BSS problem**. By leveraging our proposed criterion (10), we effectively identify pseudo features outside the true subset, thereby achieving more accurate solutions for BSS in high-correlation scenarios. This improvement constitutes an **enhancement to the solving algorithm for this NP-hard problem**, which is distinct from the objectives addressed by (robust/perturbed) sparse regression.
> > >
> > > Certainly, we can also consider best subset selection for more general objective functions, such as those incorporating robustness in (robust/perturbed) sparse regression. In Reviewer g7tW Q1, we proposed the Optimal Gradient Pursuit (OGP) scheme, which extends the Optimal Pursuit (OP) framework to general objective functions using gradient-based methods. The new metric OGP, developed under the guidance of the OP framework, remains more effective than traditional gradient pursuit approaches (see Reviewer g7tW Q1). These are indeed promising avenues for future research, but they fall beyond the scope of this paper.
> > >
> > > **Q3 (Experiment)** Thank you for your suggestion. Our definition of SNR = $20log_{10} {||X\beta||/||noise||}$ follows [1]. For the conventional SNR (calculated using $10log_{10}$), the SNR values in our rebuttal actually range from 2.5 to 7.5. We further tested scenarios with SNR values as low as 0.2–1, as shown in the figure. The algorithm using our proposed metric still demonstrates a clear phase transition advantage.
> > >
> > > [Fig: SNR Low](https://drive.google.com/file/d/1Ag3T6aTktWiaEWD7Bau5u-u4gF0Boe4i/view?usp=sharing)
> > >
> > > Due to space constraints in rebuttal, we could not elaborate in detail. In our experiments, the sparse signal is block-sparse, comprising two blocks of five adjacent non-zero entries each. Combined with the **Toeplitz covariance structure** (where **features closer in position exhibit higher correlations**), this configuration ensures:
> > > 1. High correlation features within the true subset (as stated in Theorem 1).
> > > 2. Many pseudo-features outside the true subset highly correlated with those in the true subset (as stated in Theorem 2).
> > >
> > > The observed **phase transition behavior** in the experimental results validates the theoretical superiority of our proposed criteria.
> > >
> > > We further considered more extreme high feature correlation scenarios: $corr_{ij} = \rho^{I\\{i \neq j\\}}$ with $\rho = 0.7$. As shown in the figure, the algorithm with our proposed criteria still demonstrates **significant advantages in phase transition capability**.
> > >
> > > [Fig: Corr High](https://drive.google.com/file/d/1EWsgxJ50ksYP9wdG3ad4tNuB1ggABk5Y/view?usp=sharing)
> > >
> > > [1] Block Sparse Bayesian Learning: A Diversified Scheme. NeurIPS, 2024.

---

### Official Review · Reviewer_aDws · 2025-03-13

**Overall Recommendation:** 3

**Summary:**

The paper proposes a new criterion for selecting and rejecting features in the context of the best subset selection problem. While previous methods primarily focused on the significance of individual features, the proposed approach offers the flexibility to capture interactions between features.

## update after rebuttal
Thank you for addressing my concerns. However, in light of the comments made by other reviewers, I have decided to maintain my score.

**Claims And Evidence:**

The proposed method offers the flexibility to account for the significance of individual features and the interactions among feature sets. Authors attribute the performance gains of their approach to the new modification made in the training objective. Experimental results demonstrate a significant improvement in model performance. However, I am curious about the choice of housing and superconductivity datasets for the experiments. A more comprehensive evaluation on a wider range of datasets is needed to fully validate the paper’s claims.

**Essential References Not Discussed:**

N/A

**Experimental Designs Or Analyses:**

As mentioned previously, a more comprehensive comparison across a larger set of datasets and recent baselines such as [1, 2] would provide better clarity on the usability and effectiveness of the proposed method.

[1] Cherepanova, Valeriia, et al. "A performance-driven benchmark for feature selection in tabular deep learning." Advances in Neural Information Processing Systems 36 (2023): 41956-41979.

[2] Cohen, David, et al. "Few-sample feature selection via feature manifold learning." International Conference on Machine Learning. PMLR, 2023.

**Methods And Evaluation Criteria:**

While the modification to the overall objective appears minor, the flexibility provided by the proposed approach is interesting. However, I am curious whether methods like ABESS represent the current state-of-the-art in the field. Additionally, many of the baselines used by the authors seem outdated. A comparison with more recent methods would strengthen the evaluation.

**Other Comments Or Suggestions:**

N/A

**Other Strengths And Weaknesses:**

Strength :
1. The proposed method is well-grounded in theory and offers a strong intuitive justification for the flexibility it provides over previous methods.

2. Paper is well written.

3. A simple modification in the objective function clearly leads to substantial improvements over the baselines

Weakness :
1.  I am concerned with the relevance of the proposed method with the current literature around the optimal feature selection. The author should clarify the benefits of using the proposed method over the existing state of the art or compare the performance of their method with them.

**Questions For Authors:**

N/A

**Relation To Broader Scientific Literature:**

N/A

**Theoretical Claims:**

N/A

---

> ### Author Rebuttal · Authors · 2025-04-01
>
> Thank you for your insightful feedback on our theoretical foundation and flexibility. We have incorporated further explanations and experiments accordingly, and this rebuttal will be integrated into revised paper. For the references [1-8] in the rebuttal, please refer to **Reviewer uuYf**.
>
> ---
> **Q1 (SOTA in BSS):**
> We appreciate your question, which prompted us to review the literature further. The state-of-the-art algorithm in Best Subset Selection (BSS) is ABESS, as confirmed by recent works [11]–[14], recognizing it as a leading or benchmark method.
>
> While BSS and Feature Selection (FS) are related, they are not synonymous. BSS is a subset of FS, typically in a linear framework, while FS methods like neural networks and random forests employ nonlinear models. BSS addresses the NP-hard problem (equation (2) in the original paper) with efficient polynomial-time solutions.
>
> To highlight BSS’s contributions in FS, we conducted additional experiments on larger datasets and FS tasks.
>
> ---
> **Q2 (Larger dataset, various tasks, and task-specific metrics):**
> We expanded our tests to five tasks, ten more datasets, and six task-specific metrics.
>
> 1. **Phase Transition in Extreme Scenarios:**
>    Experimental setting is detailed in Reviewer aZmj Q2, with results in: [Phase Transition](https://drive.google.com/file/d/1HY6-6XzeVTq-LUtehvQS1hZgjVvNSu7k/view).
>
>    In cases with small samples, high-dimensional features, and high noise, BSS excels in identifying the true subset, whereas other FS methods struggle.
>
> 2. **Sparse Regression Tasks on Diverse Datasets:**
>    We added three widely used BSS datasets: (1) House 16H [5], (2) Prostate.v8.egens [6-7], (3) Spectra [8]. The $R^2$ curves as a function of selected features are available here: [Fig: R2](https://drive.google.com/file/d/1v5Vz0lyVADuo2KVaUOr2ernVcSncC_t0/view?usp=sharing).
>
>    Across all datasets, the enhanced algorithms outperform the original, achieving gains equivalent to selecting ten additional features.
>
> 3. **Cross-Validation in Prediction:**
>    We evaluated BSS on six datasets using 5-fold cross-validation, where 4 folds were for training and 1 for validation. Prediction error is defined as:  $error_{pred} = \frac{1}{n} \sum_{i=1}^{n} (y_i - \hat{y}_i)^2$. Cross-validation scores are shown here:  [Fig:CV](https://drive.google.com/file/d/1xQvgSCXXpgq4nqOlkJRxJw5dUgcCbrEV/view?usp=sharing).
>
>    Enhanced algorithms exhibit superior generalization, validating the new metric’s effectiveness in predictive tasks.
>
> 4. **Column Subset Selection (CSS) in Unsupervised Learning:**
>    CSS and PCA are key dimensionality reduction methods. While PCA forms linear combinations, reducing interpretability, CSS selects important features while better preserving the dataset’s structure.
>
>    We tested eight 256×256 image datasets, using $||X - X(:,S)C||_F/||X||_F$ as the evaluation metric, with the leverage score method as a baseline. Optimal selection and deletion criteria, along with results, are here:  [Table:CCS](https://drive.google.com/file/d/1L3w-GHO9elAk4LKlhEjiDiWtR9vypUYA/view?usp=sharing).
>
>    The enhanced algorithm outperforms the original, which surpasses SVD-128, while the enhanced version consistently beats SVD-256. OP-(A)BESS achieves SOTA performance, nearing the optimal SVD bound.
>
> 5. **Line Spectrum Estimation (Complex Signal Processing):**
>    Our methods extend naturally to the complex domain. A key example is line spectrum estimation, a structured feature selection problem where features are continuous in the frequency domain:  $v(f) = [ 1, e^{-j 2\pi f}, e^{-j 2\pi 2f}, \dots, e^{-j 2\pi (N-1)f}]^T.$ This problem, crucial in modern wireless communications, involves decomposing a complex signal into its frequency components.
>
>    We tested a 128-dimensional complex signal with 20 frequency components, applying BSS on an oversampled Fourier domain. Evaluation metrics included CCDF (lower is better) for frequency estimation error and cosine similarity (higher is better) for amplitude recovery. Frequency domain visualization, radar plot, and metric performance are available here:  [Fig:LSE](https://drive.google.com/file/d/1zr1tc-udPEttYkyViwom7zZy5WbvYp4j/view?usp=sharing).
>
>    The enhanced algorithm significantly reduces frequency estimation errors and improves correlation. OP-(A)BESS and CoSaOP achieve perfect estimation, reinforcing our metric’s advantage in highly correlated feature settings.
>
> ---
> **References:**
>
> [11] Wang, Zezhi, et al. "skscope: Fast Sparsity-Constrained Optimization in Python." JMLR (2024).
>
> [12] Roy, Saptarshi, et al. "On the Computational Complexity of Private High-dimensional Model Selection." NeurIPS (2024).
>
> [13] Lin, Zhaotong, et al. "A robust cis-Mendelian randomization method with application to drug target discovery." Nature Communications (2024).
> [14] Gregory C. Reinsel, et al. Multivariate Reduced-Rank Regression: Theory, Methods and Applications. Springer (2022).

---

### Official Review · Reviewer_uuYf · 2025-03-18

**Overall Recommendation:** 4

**Summary:**

The paper presents two novel criteria for feature selection, which are refinements on well studied approaches for identifying features which maximally improve (or reduce) prediction accuracy. By more rigorously considering the impact of features selected as a subset, rather than just individually, similar efficiency guarantees remain while improvements and learning are made.

**Claims And Evidence:**

Yes, all claims are supported with theoretical proofs and experimental results.

**Essential References Not Discussed:**

na

**Experimental Designs Or Analyses:**

I reviewed the experimental results. I think a more thorough discussion of CoSaMP's failure is needed, both in terms of why the failure occurs and more experimental validation that this result is not in error.

**Methods And Evaluation Criteria:**

Yes.

**Other Comments Or Suggestions:**

na

**Other Strengths And Weaknesses:**

The paper is very well written, with helpful illustrations to hammer home the nuance in the idea for improving prior formalizations of the problem. Performance gains are further made abundantly clear by the provided experimental results. Moreover, the generality of these results is what's most intriguing to me--the methods take a fundamentally new approach to well studied solutions.

**Questions For Authors:**

How does this paper differ from Tohidi et al 2025? The submodularity and matching pursuit approach their seem to overlap significantly with this work and she be discussed in more depth.

**Relation To Broader Scientific Literature:**

Yes, the authors well situate their results as they compare to prior feature selection methods.

**Theoretical Claims:**

I verified the proofs of Theorem 3.3 and 3.11 in the appendix.

---

> ### Author Rebuttal · Authors · 2025-04-01
>
> We sincerely appreciate your deep understanding and kind words on our work. We have carefully addressed your comments below and will incorporate this rebuttal into the revised version of the paper.
>
> ---
> **Q1 (Why CoSaMP Fails):**
>
> Thank you for your question. We implemented Algorithm 1 of CoSaMP [1] strictly and tested it alongside several example codes from MathWorks, obtaining consistent results. We will now provide a deeper analysis of the reasons behind CoSaMP's inefficiency in sparse regression tasks.
>
> As discussed in Appendix M, CoSaMP iteratively (1) selects $2K$ features, (2) solves a least squares problem on a large subset, and (3) prunes to $K$ coefficients. However, high feature correlation can cause significant errors in the final estimate from steps (2) and (3). We visualize the impact of feature correlation on CoSaMP's iterative process here: [Fig: CoSaMP Visualization 1](https://drive.google.com/file/d/1Zs1C0Bp3NVnu0RqriajYSfJkcHyeWNYI/view?usp=sharing).
>
> As shown, on a regression dataset with highly correlated features, the pruned support set’s direct coefficients (column 2) differ significantly from those after least squares estimation (column 3), with substantial residuals. CoSaMP, lacking least squares refinement, fails as the residuals grow with each iteration due to high feature correlation. This is evident in the residual curve evolution in (a): [Fig: Residual Curve](https://drive.google.com/file/d/1xp34dLCafqFV3tSvEHjkvsqEFTx_ehoM/view?usp=sharing).
>
> In contrast, when features are weakly correlated (as in the Audioset [Fig: Audioset Example](https://drive.google.com/file/d/1vJiWa-jqo4NNSPAbRU2Ew6XXSBmDHyY7/view?usp=share_link)), the coefficients and residuals after pruning the large support set (column 2) and performing least squares (column 3) are nearly identical, leading to algorithm convergence, as shown in curve (b): [Fig: Residual Curve](https://drive.google.com/file/d/1xp34dLCafqFV3tSvEHjkvsqEFTx_ehoM/view?usp=sharing).
>
> In summary, as noted in [1], CoSaMP's theoretical guarantees rely on weak feature correlation, leading to failure when this assumption is violated. In contrast, our CoSaOP algorithm remains effective. We further establish its theoretical foundation (**see Reviewer aZmj, Q1, Theorems 1, 2**), demonstrating how the new criteria enable algorithms that overcome high feature correlation challenges.
>
> ---
> **Q2 (Difference from Tohidi et al 2025):**
>
> Thank you for your question.
> Tohidi et al. (2025) use submodularity and preconditioning, applicable only to the selection process. Our contribution, however, re-examines the foundational criteria in best subset selection from an optimization perspective. By modeling feature independence and interactions through block coordinate descent, we clarified the optimization essence of classical criteria and proposed a unified feature selection and elimination model. Using forward and backward matrix inversion, we derived new explicit criteria, providing a foundation for future algorithm design in best subset selection.
>
> In this paper and rebuttal, we analyzed: (1) the criteria's theory under high feature correlations, (2) complexity and convergence, (3) empirical performance gains, and (4) performance across various tasks and metrics. These findings highlight the potential to relax theoretical assumptions and enable future engineering applications.
>
> Moreover, the optimal pursuit idea can be extended to other greedy metrics and algorithms, opening new directions for further research (**see Reviewer g7tw Q1**).
>
> ---
> **References:**
>
> [1] Needell D, Tropp J A. CoSaMP: Iterative signal recovery from incomplete and inaccurate samples[J]. *Applied and computational harmonic analysis*, 2009, 26(3): 301-321.
>
> [2] Hastie, T., Tibshirani, R. and Friedman, J. (2017) *The Elements of Statistical Learning: Data Mining, Inference, and Prediction*. 2nd Edition, Springer, Berlin.
>
> [3] Belhadji A, Bardenet R, Chainais P. A determinantal point process for column subset selection[J]. *Journal of machine learning research*, 2020, 21(197): 1-62.
>
> [4] Blumensath T, Davies M E. Gradient pursuits[J]. *IEEE Transactions on Signal Processing*, 2008, 56(6): 2370-2382.
>
> [5] [OpenML Data](https://www.openml.org/search?type=data&sort=runs&id=574&status=active)
>
> [6] Lin Z, Pan W. A robust cis-Mendelian randomization method with application to drug target discovery[J]. *Nature communications*, 2024, 15(1): 6072.
>
> [7] Hastie, T., Tibshirani, R. and Friedman, J. (2017) *The Elements of Statistical Learning: Data Mining, Inference, and Prediction*. 2nd Edition, Springer, Berlin.
>
> [8] MATLAB and Statistics and Machine Learning Toolbox, "Spectra Data," The MathWorks, Inc.

---

### Decision · Program_Chairs · 2025-05-01

**Decision:**

Accept (poster)

**Comment:**

This paper proposed two criteria for feature selection and feature elimination for the best subset selection problem, which are motivated from optimization perspective and incorporated into various subset selection algorithms. Convergence guarantees for CoSaOP is established. Reviewers raised some concerns regarding the assumptions of the theoretical results, the incremental contribution to existing work, outdated baselines, and the simulation data in experimental design. Authors made great efforts on addressing these concerns with more theoretical results and experiments. However, some critical concerns remain unsolved: the presented theoretical results are not explicitly supporting for strong correlation even though the additionally presented experimental results empirically demonstrated that; outdated baselines can be improved by including more recent works. As a result, weak accept is recommended.